**Dynamics-based estimates of decline trend with fine temporal variations in China's PM$_{2.5}$ emissions**

**Zhen Peng[1]†, Lili Lei[1,2]†, Zhe-Min Tan[1,2]*, Meigen Zhang[3]*, Aijun Ding[1] and Xingxia Kou[4]**

[1]School of Atmospheric Sciences, Nanjing University, Nanjing 210093, China

[2]Key Laboratory of Mesoscale Severe Weather, Ministry of Education, Nanjing University, Nanjing 210093, China

[3]State Key Laboratory of Atmospheric Boundary Layer Physics and Atmospheric Chemistry, Institute of Atmospheric Physics, Chinese Academy of Sciences, Beijing 100029, China

[4]Institute of Urban Meteorology, China Meteorological Administration, Beijing 100089, China

Corresponding author: Zhe-Min Tan (zmtan@nju.edu.cn) and Meigen Zhang (mgzhang@mail.iap.ac.cn)

**Abstract**

Timely, continuous, and dynamics-based estimates of PM$_{2.5}$ emissions with a high temporal resolution can be objectively and optimally obtained by assimilating observed surface PM$_{2.5}$ concentrations using flow-dependent error statistics. The annual dynamics-based estimates of PM$_{2.5}$ emission averaged over mainland China for years 2016-2020 without biomass burning emissions are 7.66, 7.40, 7.02, 6.62 and 6.38 Tg, respectively, which are very closed to the values of MEIC. Annual PM$_{2.5}$ emissions in China have consistently decreased of approximately 3% to 5% from 2017 to 2020. Significant PM$_{2.5}$ emission reductions occurred frequently in regions with large PM$_{2.5}$ emissions. COVID-19 could cause a significant reduction of PM$_{2.5}$ emissions in the north China plain and northeast of China in 2020. The magnitudes of PM$_{2.5}$ emissions were greater in the winter than in the summer. PM$_{2.5}$ emissions show an obvious diurnal variation that varies significantly with the season and urban population. Compared to the diurnal variations of PM$_{2.5}$

emission fractions estimated based on diurnal variation profiles from US and EU, the estimated PM$_{2.5}$ emission fractions are 1.25% larger during the evening, the morning peak is 0.57% smaller in winter and 1.05% larger in summer, and the evening peak is 0.83% smaller. Improved representations of PM$_{2.5}$ emissions across time scales can benefit emission inventory, regulation policy and emission trading schemes, particularly for especially for high temporal resolution air quality forecasting and policy response to severe haze pollutions or rare human events with significant socioeconomic impacts.

## 1. Introduction

Anthropogenic emissions have imposed essential influences on the earth system, from hourly air quality and human health to long-time climate and environment. To reduce anthropogenic emissions, the Chinese government has enforced the Clean Air Action (2013) since 2013. Studies to date that evaluated the emission controls and understood the climate responses from emission reductions often have used either a fixed meteorology with emission changes or *vice versa* (Li et al., 2019a; Li et al., 2021, Zhai et al., 2021). Estimated emissions from empirical extrapolation were commonly applied to analyze the meteorological-chemical mechanisms and associated social-economic impacts from occasional events like the 2015 China Victory Day Parade and Coronavirus Disease 2019 (COVID-19) pandemic (Wang et al., 2017; Liu et al., 2020; Huang et al., 2020; Zhu et al., 2021). But to better understand both long-term and short-term influences from emission changes, the continuous, up-to-date, and high temporal-/spatial-resolution emission estimates with coherent interactions of meteorology and emission changes are needed.

The complex contributions from energy production, industrial processes, transportation, and residential consumptions have imposed great challenges to accurately estimate the emissions. The emission inventories created by the traditional bottom-up techniques were typically outdated from the present day due to the lack of accurate and timely statistics, and often with coarse temporal resolutions from monthly to annual (Zhang et al., 2009; Li et al., 2014; Janssens-Maenhout et al., 2015; Zheng et al., 2018). Alternatively, update-to-date emission estimates with high temporal-spatial resolutions could be provided by top-down techniques (Miyazaki et al., 2017), but most emissions estimated by top-down techniques were intermittent and analyzed at

monthly scale or longer longer (Zhang et al., 2016; Jiang et al., 2017; Qu et al., 2017; Cao et al., 2018; Müller et al., 2018; Chen et al., 2019; Li et al., 2019b; Miyazaki et al., 2020). Moreover, emissions updated by the top-down techniques based on satellite observations could be insufficient to capture realistic near-surface characteristics (Li et al., 2019b; Liu et al., 2011; Choi et al., 2020).

Given the development of observation networks and advanced data assimilation strategies, timely and dynamics-based emission estimates with high temporal resolution can be achieved by harmonically constraining the atmospheric-chemical model with dense observations of trace gas compounds through an optimal assimilation methodology. The ensemble Kaman smoother (EnKS) (Whitaker et al., 2002; Peters et al., 2007; Peng et al., 2015), as a four-dimensional (4D) assimilation algorithm, makes use of chemical observations from past to future to provide an optimal estimate of source emissions, and it can capture the "error of the day" and construct fine emission characteristics with high temporal-spatial resolutions by using short-term ensemble forecasts (Kalnay, 2002). Since 2013, the fine particulate matter pollution (PM$_{2.5}$, particles smaller than 2.5 μm in diameter) as the most urgent threat to public health has been persistently decreased, and ground-based observations of PM$_{2.5}$ have been progressively increased (Huang et al., 2018). Thus by harmonically assimilating dense surface PM$_{2.5}$ observations into an atmospheric-chemical model through an EnKS, hourly estimates of PM$_{2.5}$ emission that were continuously cycled for years 2016-2020 are presented in this study.

The timely estimated emissions can provide guidance for emission inventories that usually have time lags and emission trading schemes that often require up-to-date source emissions. Based on the dynamics-based estimated emissions with harmonic combination of the model and observations, better evaluation of the emission controls and more comprehensive understanding of the consequent climate responses can be obtained. The high temporal-resolution estimated emissions can reveal features of emissions that are absent from the traditional ones with coarse temporal resolutions. Moreover, the timely and dynamics-based emission estimates with high temporal resolution are essential for regional air quality modeling, especially for the occurrence of severe haze pollutions associated with timely evaluation for the impact on public health (Attri et al., 2001; Wang et al., 2014; Ji et al., 2018; Wang et al., 2020; Liu et al., 2021) and events that lead to large changes of emissions and significant socioeconomic impacts such as the COVID-19 pandemic (Huang et al., 2020; Le et al., 2020).

## 2. Data assimilation and experimental design

The estimate of PM$_{2.5}$ emission can be successfully constrained by the PM$_{2.5}$ concentration observations through an ensemble Kalman filter (EnKF; Peng et al., 2017, 2018, 2020). For a retrospective 'reanalysis' mode here, all available PM$_{2.5}$ concentration observations, including those data collected after the analysis time, can be used. Thus a EnKS, a direct generalization of the EnKF, is applied to incorporate PM$_{2.5}$ concentration observations both before and after the analysis time, aiming to provide an optimal estimate of the PM$_{2.5}$ emission. In simple words, The emissions are updated by current and future observations though EnKS, while the concentrations are updated by current observations though EnKF. Detailed procedures of the EnKS are described in section 2.1.

## 2.1 An ensemble Kalman smoother to update the source emission

The ensemble priors of source emissions $\mathbf{e}^f$ is created by multiplying a scaling factor $\boldsymbol{\lambda}^f$ to the prescribed emission $\mathbf{e}^{\mathbf{P}}$ (Peng et al., 2017, 2018, 2020), where the superscript $f$ denotes priors. Given a time-invariant $\mathbf{e}^{\mathbf{P}}$, the update of $\mathbf{e}^f$ is equivalent to the update of $\boldsymbol{\lambda}^f$. Due to a time lag, the prior scaling factor at time $t$-1 ($\boldsymbol{\lambda}^f_{t-1}$) is updated by chemical observations at time $t$ ($\mathbf{y}^c_t$). At time $t$-1, the prior scaling factor for the $i^{\text{th}}$ member is written as

$$\boldsymbol{\lambda}^f_{i,t-1} = \frac{1}{M}\left[\left(\beta\frac{\mathbf{c}^f_{i,t-1}}{\overline{\mathbf{c}}^f_{t-1}}+1-\beta\right)+\sum_{j=t-M}^{t-2}\boldsymbol{\lambda}^a_{i,j|j+1:t-1}\right]. \tag{1}$$

The first term is the concentration ratio given by the prior of the chemical fields ($\mathbf{c}^f_{i,t-1}$) normalized by the ensemble mean ($\overline{\mathbf{c}}^f_{t-1}$), where $\beta$ is an inflation factor used to compensate the insufficient ensemble spread (Peng et al., 2017). Through using the concentration ratio, each ensemble member of the source emissions naturally has the spatial correlations given by the chemical fields. The second term is the mean of the posterior scaling factors at previous assimilation cycles, where the superscript $a$ denotes posteriors, $M$ is the length of smoothing, and the subscript $j$+1:$t$-1 indicates that the scaling factor at time $j$ is updated by future observations from $j$+1 to $t$-1. The assimilation of future observations will be described below.

The ensemble square-root filter (EnSRF) (Peng et al., 2017) is used to update $\lambda_{t-1}^{f}$ by

assimilating $\mathbf{y}_{t}^{c}$. For the scaling factor at time $t$-1, posterior ensemble mean is given by

$$\overline{\boldsymbol{\lambda}}_{t-1}^{a} = \overline{\boldsymbol{\lambda}}_{t-1}^{f} + \rho \circ \mathbf{P}_{t-1,t}^{ec} \mathbf{H}_{t}^{c\mathbf{T}} \left( \mathbf{H}_{t}^{c} \mathbf{P}_{t}^{c} \mathbf{H}_{t}^{c\mathbf{T}} + \mathbf{R}_{t}^{c} \right)^{-1} \left( \mathbf{y}_{t}^{c} - H_{t}^{c} \overline{\mathbf{c}}_{t}^{f} \right),$$ (2)


and posterior ensemble perturbations are given by
$$\boldsymbol{\lambda}_{i,t-1}^{'a} = \boldsymbol{\lambda}_{i,t-1}^{'f} - \rho \circ \mathbf{P}_{t-1,t}^{ec} \mathbf{H}_{t}^{c\mathbf{T}} \left[ \left( \sqrt{\mathbf{H}_{t}^{c}\mathbf{P}_{t}^{c}\mathbf{H}_{t}^{c\mathbf{T}} + \mathbf{R}_{t}^{c}} \right)^{-1} \right]^{\mathbf{T}} \left[ \sqrt{\left( \mathbf{H}_{t}^{c}\mathbf{P}_{t}^{c}\mathbf{H}_{t}^{c\mathbf{T}} + \mathbf{R}_{t}^{c} \right)} + \sqrt{\mathbf{R}_{t}^{c}} \right]^{-1} \mathbf{H}_{t}^{c} \boldsymbol{\lambda}_{i,t-1}^{'f} ,$$

(3)

where $\mathbf{P}_{t-1,t}^{ec}$ denotes the background error covariance matrix of $\lambda_{t-1}^{f}$ and $\mathbf{c}_{t}^{f}$, $\mathbf{P}_{t}^{c}$ indicates the
background error covariance matrix of $\mathbf{c}_{t}^{f}$, $H_{t}^{c}$, $\mathbf{H}_{t}^{c}$ and $\mathbf{R}_{t}^{c}$ are the observation forward operator,
Jacobian matrix and observation error covariance matrix of the chemical fields at time $t$, $\rho$ is the
localization matrix and $\circ$ denotes the Schur (elementwise) product.

By applying the ensemble Kalman smoother (EnKS) (Whitaker et al., 2002; Peters et al.,

2007), the chemical observation $\mathbf{y}_{t}^{c}$ is also assimilated to update the posterior scaling factor at
previous assimilation cycles $j\left( j = t - K, \ldots, t - 2 \right)$. After assimilating the future chemical
observation at time $t$, posterior ensemble mean of the scaling factor at $j$ is given by

$$\overline{\boldsymbol{\lambda}}_{j|j+1:t}^{a} = \overline{\boldsymbol{\lambda}}_{j|j+1:t-1}^{a} + \rho \circ \mathbf{P}_{j|j+1:t-1,t}^{ec} \mathbf{H}_{t}^{c\mathbf{T}} \left( \mathbf{H}_{t}^{c} \mathbf{P}_{t}^{c} \mathbf{H}_{t}^{c\mathbf{T}} + \mathbf{R}_{t}^{c} \right)^{-1} \left( \mathbf{y}_{t}^{c} - H_{t}^{c} \overline{\mathbf{c}}_{t}^{f} \right),$$ (4)


and posterior ensemble perturbations are given by

$$\boldsymbol{\lambda}_{i,j|j+1:t}^{'a} = \boldsymbol{\lambda}_{i,j|j+1:t-1}^{'a} -$$

$$\rho \circ \mathbf{P}_{j|j+1:t-1,t}^{ec} \mathbf{H}_{t}^{c\mathbf{T}} \left[ \left( \sqrt{\mathbf{H}_{t}^{c}\mathbf{P}_{t}^{c}\mathbf{H}_{t}^{c\mathbf{T}} + \mathbf{R}_{t}^{c}} \right)^{-1} \right]^{\mathbf{T}} \left[ \sqrt{\left( \mathbf{H}_{t}^{c}\mathbf{P}_{t}^{c}\mathbf{H}_{t}^{c\mathbf{T}} + \mathbf{R}_{t}^{c} \right)} + \sqrt{\mathbf{R}_{t}^{c}} \right]^{-1} \mathbf{H}_{t}^{c} \boldsymbol{\lambda}_{i,t-1}^{'f}$$

, (5)

where $\mathbf{P}_{j|j+1:t-1,t}^{ec}$ denotes the background error covariance matrix of $\lambda_{j|j+1:t-1}^{a}$ and $\mathbf{c}_{t}^{f}$. After (2)-(5),
the updated $\lambda_{j|j+1:t}^{a}, j\left( j = t - M + 1, \ldots, t - 1 \right)$ will be used to construct the prior scaling factor at next
time $t$+1 (1).
As a Monte Carlo approach, the EnKS uses the forecast-analysis error covariances based
on ensemble forecasts / analyses to compute the Kalman gain matrix with time lags, to incorporate
observations from the past to the future. The first iteration of EnKS is equivalent to EnKF that
assimilates observations up to the analysis time. The following iterations of EnKS assimilate
observations in the future to update the state at the analysis time. The hourly forecasts of $PM_{2.5}$
concentration from the cycling assimilation experiment matched the independent observed
quantities (Figure 1). Therefore, the ability of EnKS to retrieve the source emissions has been
demonstrated. Previous studies also showed that simulations forced by the posterior emissions
could produce improved forecasts for $PM_{2.5}$, $SO_2$, and $NO_2$ than those with a priori emissions
(Peng et al., 2020).

**2.2 WRF-Chem model, observations and emissions**

To simulate the transport of aerosol and chemical species, the WRF-Chem model version
3.6.1 (Grell et al., 2005) that has the meteorological and chemical components fully coupled is
used. The model parameterization schemes follow Peng et al. (2017). Figure 2 shows the model
domain that covers most east Asia regions. Horizontal grid spacing is 45 km with 57 vertical levels
and model top at 10 hPa.
Experiments are conducted for each year from 2016 to 2020 separately. The 6-h
meteorological observations, including all in-situ observations and cloud motion vectors from the
National Centers for Environmental Prediction (NCEP) Global Data Assimilation System (GDAS;
http://www.emc.ncep.noaa.gov/mmb/data_processing/prepbufr.doc/table_2.htm), are assimilated
every 6 h. The hourly observed chemical quantities, which contain $PM_{10}$, $PM_{2.5}$, $SO_2$, $NO_2$, $O_3$,
and CO from the Ministry of Ecology and Environment of China (https://aqicn.org/map/china/cn/),
are assimilated every hour. Figure 2 shows the assimilated chemical observation network, which
has 560 randomly chosen stations from 1576 stations in total. The thinning of observations is
applied to avoid correlated errors of observations. The spatial autocorrelation of the thinning of
observations is close to the original observations (Peng et al., 2017). The observation priors are
computed by the "observer" portion of the Grid-point Statistical Interpolation system (GSI) (Kleist
et al., 2009).
The hourly and time-invariantly prescribed anthropogenic emissions are obtained from the
EDGAR-HTAP (Emission Database for Global Atmospheric Research for Hemispheric Transport
of Air Pollution v2.2) v2.2 inventory (Janssens-Maenhout et al., 2015), in which the Chinese
emissions are derived from MEIC in 2010 (Lei et al., 2011; Li et al., 2014). Natural emissions,
including the biogenic (Guenther et al., 1995), dust (Ginoux et al., 2001), dimethyl sulfide and sea
salt emissions (Chin et al., 2000), are computed online.

**2.3 Assimilation and ensemble configurations**

The $PM_{2.5}$ emission directly gives the primary $PM_{2.5}$, and then the primary $PM_{2.5}$ along
with other precursor emissions could contribute to the secondary $PM_{2.5}$. The observations of $PM_{2.5}$
concentrations that contain both primary and secondary $PM_{2.5}$, are used to constrain the $PM_{2.5}$
emission through data assimilation. Thus the correlations between the concentration observations
and source emissions might be contaminated by the secondary $PM_{2.5}$. Since the secondary
formation process can be captured by the WRF-Chem model, the impact of the secondary $PM_{2.5}$
is indirectly considered. The detailed updated state variables with the according observations
follow Peng et al. (2018). The concentrations and emissions of $PM_{2.5}$, $NH_3$, and $PM_{2.5}$ precursors
that have observations ($SO_2$ and NO), are updated by the observed quantities, respectively, but the
VOC that are also $PM_{2.5}$ precursors are not updated due to the lack of direct and limited
observations.. One possible way to untangle the impact of secondary $PM_{2.5}$ on the estimates of
$PM_{2.5}$ emission is to jointly estimate the source emission, primary and secondary $PM_{2.5}$ given the
concentration observations.
The National Oceanic and Atmospheric Administration (NOAA) operational EnKF system
(https://dtcenter.ucar.edu/com-GSI/users/docs/users_guide/GSIUserGuide_v3.7.pdf), which is an
EnSRF and modified with the EnKS feature, is used to assimilate the observations. Ensemble size
is set to 50. To combat the sampling error resulted from a limited ensemble size, covariance
localization and inflation are applied. The Gaspari and Cohn (GC) (1999) function with a length
scale of 675 km is used to localize the impact of observations and mitigate the spurious error
correlations between observations and state variables. The constant multiplicative posterior
inflation (Whitaker and Hamill 2012) with coefficients 1.12 for all meteorological and chemical
variables is applied to enlarge the ensemble spread. The inflation $\beta$ for advancing the scale factor
is 1.2. The smoothing length *M* for source emissions is 4, and the EnKS lagged length *K* is 6. The
larger the K value, the more future observations are assimilated to constrain the current emission
estimate. But the sample estimated temporal correlations could be contaminated by sampling errors
and model errors, especially with increased lagged times. Thus, there is a tradeoff between the
amount of future observations and accuracy of sample estimated temporal correlations. The choice
of K (=6) is determined by sensitivity experiments.
At 0000 UTC 26 December of previous year, ensemble initial conditions (ICs) of the
meteorological fields are generated by adding random perturbations that sample the static
background error covariances (Barker et al., 2012) on the NCEP FNL (Final) analyses (Torn et al.,
2006). Ensemble ICs of the chemical fields are 0, and source emissions of each ensemble member
are adopted from the EDGAR-HTAP v2.2 inventory with random perturbations of mean 0 and
variances of 10% of the emission values. Hourly ensemble lateral boundary conditions (LBCs) are
generated using the same fixed-covariance perturbation technique as the ensemble ICs. After 6-d
spin up, ensemble data assimilation experiments start cycling for each year.

## 3. $PM_{2.5}$ emission for years 2016-2020

Starting from the time-invariant source emission PR2010 (Janssens-Maenhout et al., 2015),
the dynamics-based estimates of the $PM_{2.5}$ emissions are obtained, which include both the
contributions of the anthropogenic and biomass burning emissions. The mean annual $PM_{2.5}$
emissions from biomass burning in China (2003~2017) was 0.51 Tg (Yin et al., 2019). The annual
dynamics-based estimates of $PM_{2.5}$ emission (DEPE) averaged over mainland China for years
2016-2020 without biomass burning emissions are 7.66, 7.40, 7.02, 6.62 and 6.38 Tg, respectively.
The values from the Multi-resolution Emission Inventory (MEIC; Zheng et al., 2018) that does not
consider the contributions of biomass burning emissions, are 8.10, 7.60, 6.70, 6.38 and 6.04 Tg,
respectively. Thus the annual DEPE are very closed to the values of MEIC. From year 2017 to
2020, the estimated annual $PM_{2.5}$ emissions are reduced 3.4%, 8.4%, 13.6% and 16.7%
respectively compared to that of year 2016. There has been 3%-5% persistent reduction of annual
$PM_{2.5}$ emission from year 2017 to 2020, which demonstrates the effectiveness of China's Clean
Air Action (2013) implemented since 2013 and China Blue Sky Defense War Plan (2018) enforced
since 2018 with strengthened industrial emission standards, phased out outdated industrial
capacities, promoted clean fuels in residential sector and so on (Zhang et al., 2019).
The monthly DEPE show reduction of $PM_{2.5}$ emission nearly in each month from years
2016 to 2020 (Figure 3a), which further demonstates the effectiveness of China's national plan.
Compared to year 2016, both the reduction amount and reduction ratio of $PM_{2.5}$ emission are more
prominent for February, March, June-September, and November than the other months (Figure3b).
Given larger magnitudes of $PM_{2.5}$ emission in winter than in summer, emission controls with a
focus from October to May should be considered in the design of future clean air actions in China,
since total $PM_{2.5}$ emission during this period accounts for approximate 75% annual amount. Spatial
distributions of the changes of $PM_{2.5}$ emission from year 2017 to 2020 compared to year 2016
show significant decreases occurred at Beijing-Tianjin-Hebei region (BTH), Yangtze River Delta
region (YRD), Pearl River Delta region (PRD) and Sichuan-Chongqing Region (SCR), especially
for years 2019-2020 (Figure 4). From year 2016 to 2020, BTH, YRD and SRC have larger
reductions of $PM_{2.5}$ emission than PRD, but SCR has larger reduction ratio compared to year 2016
than BTH and YRD (Figure 5). Therefore, BTH and YRD have more potentials for $PM_{2.5}$ emission
controls than PRD and SCR, which can give a guidance for future clean air actions. More
specifically, most provinces have $PM_{2.5}$ emission reduction from year 2016 to 2020, and the
reduction ratios generally increase from year 2017 to 2020 (Table 1), which confirms continuous
and effective emission controls from Clean Air Action to Blue Sky Defense War Plan in China.
The monthly DEPE also demonstates the effectiveness of strict implementations of emission
reduction policies in China, such as the coal ban for residential heating since the 2017-2018 winter.
There was a sharp change of $PM_{2.5}$ emission, from increase in 2017 to decrease in 2018. As shown
by Figure 6, spatial distributions of the changes of $PM_{2.5}$ emissions in December compared to
November in 2017 show obvious increases in most China. However, the changes in 2018 show
significant decreases in areas of Beijing, Tianjin, Hebei, Shanxi, Henan and Anhui provinces due
to the implementation of the coal ban.
Despites the trend in $PM_{2.5}$ emissions from year 2016 to 2020, the DEPE of year 2016 has
similar monthly distributions to MEIC2016-2020 in general (Figure 3a). MEIC has a "Pan-shape"
monthly distribution with nearly time-invariant $PM_{2.5}$ emissions from April to October. This
seasonal dependence of emissions is mainly contributed by the variations of residential energy use,
which are empirically dependent on coarse monthly mean temperature intervals and thus cannot
reflect the realistic monthly variations (Streets et al., 2003; Li et al., 2017). The centralized heating
system in North China has a fixed date of turning-on and turning-off during each heating season.
Therefore, a sudden raise of emissions from October to November and a sudden drop of emissions
from March to April are shown. But the turning-on and turning-off date are variable in different
regions, which imposes a smoothing impact on the emissions. However, the DEPE yet shows a
"V-shape" monthly distribution, with the minimum occurring in August. The estimated PM$_{2.5}$
emission is 11.8% higher than MEIC2016 in April but 12.1% lower than MEIC2016 in August,
and these different monthly distributions can influence the consequent climate responses including
the radiative forcing and energy budget (Yang et al., 2020) and also impact the health issues (Liu
et al., 2018). Moreover, monthly fractions of the DEPE are consistent cross years (Figure 3c). The
absence of interannual variations of monthly PM$_{2.5}$ emission fraction provides basis for previous
studies that follow the same monthly changes of source emissions from different years (Zhang et
al., 2009; Zheng et al., 2020, 2021). Monthly allocations of PM$_{2.5}$ emission can be directly and
objectively obtained given an estimated total annual amount based on the estimated monthly
fractions of DEPE, which is valuable for emission inventory, air quality simulation, and potentially
applications for future scenarios due to more accurate month fractions of DEPE. Since the hourly
priors of PM$_{2.5}$ concentrations from the cycling assimilation for optimally estimating PM$_{2.5}$
emission fit to the observed PM$_{2.5}$ quantities (Figure 1), the monthly DEPE provides more realistic
monthly fluctuations than the empirical estimate.
**4. Diurnal variations of PM$_{2.5}$ emission**
The DEPE with high temporal-resolution given the time-invariant prior PR2010 can reveal
features that are unable to represent in the commonly used emission estimates. Although the prior
PR2010 has no diurnal variations, hourly posteriors of PM$_{2.5}$ emission provide the first objectively
estimated diurnal variations for different seasons for years 2016-2020. However, these estimated
diurnal variations include the contributions of the time-varying boundary layer. An observing
system simulation experiment (OSSE) is performed to investigate the effects of the boundary layer.
Details of this OSSE are presented in Appendix. The results indicate that the magnitude of
posterior PM$_{2.5}$ emission from the OSSE is closer to the true emission than the prior. Since we
have hourly assimilated observations to simultaneously update the chemical concentrations and
source emissions, the impacts of time-varying boundary layer on the posterior PM$_{2.5}$ emissions are
limited (Figures S1). A little larger estimated PM$_{2.5}$ emission fractions occurred in the morning
and smaller estimated PM$_{2.5}$ emission fractions occurred in the afternoon, comparing to the time-
invariant true emission. Nevertheless, the influences of time-varying boundary layer are still
important to PM$_{2.5}$ emission estimates. To statistically present the diurnal variations, the fractions
of hourly PM$_{2.5}$ emissions divided by the daily amount are averaged over different years and
regions after excluding the impacts of time-varying boundary layer (Figures 7 and 8, and Table 2).
The diurnal variations of $PM_{2.5}$ emission are critical for understanding the mechanisms of $PM_{2.5}$
formation and evolution and are also essential for $PM_{2.5}$ simulation and forecast.
Five-year mean diurnal variations of the estimated $PM_{2.5}$ emission fraction for mainland
China show that despite the monthly variations of $PM_{2.5}$ emission, the diurnal-variation fractions
for November, December, January and February are similar, while those for June, July and August
are similar (Figure 7a). There are stronger diurnal variations of $PM_{2.5}$ emission in summer than in
winter, which are represented by larger $PM_{2.5}$ emission fractions during morning and less $PM_{2.5}$
emission fractions during evening. The diurnal variations of $PM_{2.5}$ emission from March to May
gradually transform from the patterns of winter to those of summer, and *vice versa* for the diurnal
variations of $PM_{2.5}$ emission from September to November. The monthly changes of diurnal
variations of $PM_{2.5}$ emission are consistent with the seasonal dependence, since monthly variations
of $PM_{2.5}$ emission are mainly related to the variations of residential consumptions (Li et al., 2017)
in which the space-heating has nearly no diurnal variations and then larger $PM_{2.5}$ emissions during
winter lead to reduced diurnal variations than summer. Similar to the monthly fractions of
estimated $PM_{2.5}$ emission for mainland China, diurnal variations of $PM_{2.5}$ emission fraction are
consistent cross years for a given month (Figure 8). Table 2 gives five-year mean diurnal variations
of the estimated $PM_{2.5}$ emission fraction for each month. Based on these high-resolution diurnal-
variation fractions, hourly estimates of $PM_{2.5}$ emission can be objectively obtained for a given
monthly estimated $PM_{2.5}$ emission.
Despite the high temporal resolution, the DEPE also has the ability to analyze diurnal
variations for specific cities. The monthly changes of diurnal variations of $PM_{2.5}$ emission
estimated for megacities with urban populations larger than 5 million and non-megacities with
urban populations smaller than 5 million (Notice of the State Council on Adjusting the Standards
for Categorizing City Sizes, 2014) are consistent with those estimated from mainland China
(Figure 7). Compared to the diurnal variations of $PM_{2.5}$ emission estimated for mainland China,
the megacities have stronger diurnal variations, while the non-megacities have weaker diurnal
variations. These detailed descriptions of $PM_{2.5}$ emission that are usually absent in common
emission estimates can be essential for $PM_{2.5}$ simulation, especially for providing timely and
realistic guidance for severe haze events.

There has been lack of local measurements for diurnal variations and widely adopted

diurnal variation profiles of $PM_{2.5}$ emission in China. Compared to the diurnal variations of $PM_{2.5}$
emission fractions estimated based on diurnal variation profiles from US and EU (Wang et al.,
2010; Du et al., 2020), the estimated $PM_{2.5}$ emission fractions are 1.25% larger during the evening, ,
which greatly changes the diurnal variations of DEPE. The noon and evening peaks estimated from
DEPE have smaller $PM_{2.5}$ emission fractions, with mean underestimations of $PM_{2.5}$ emission
fraction of0.40% and 0.83% for noon peak and evening peak respectively (Figures 7a and 9). In
fact, the smaller evening peaks of Wang et al. (2010) occurred in November, December, January,
February and March, while they are almost indistinct from April to October, similar to that from
DEPE. The morning peak of Wang et al. (2010) is similar to that of DEPE for spring and fall, but
the former overestimates $PM_{2.5}$ emission fraction of 0.57% for winter while underestimates $PM_{2.5}$
emission fraction of 1.05% for summer. Due to the overestimated peaks, diurnal variations of
Wang et al.(2010) have sharper appearance rate for morning peak and disappearance rate for
evening peak. Compared to the diurnal variations based on diurnal variation profiles from ES and
EU (Wang et al., 2010), the diurnal variations of the DEPE are constrained by the atmospheric-
chemical model and observed $PM_{2.5}$ concentrations, which can objectively determine the diurnal
variations of $PM_{2.5}$ emission for specific regions and seasons.
**5. Impact of COVID-19 on $PM_{2.5}$ emissions**

The abrupt outbreak of the COVID-19 pandemic has produced dramatically socioeconomic

impacts in China. To prevent the virus spread, a lockdown was first implemented on 23 January
2020 in Wuhan, Hubei province, and subsequently the national lockdown has been enforced in
China (Liu et al., 2020; Huang et al., 2020; Zhu et al., 2021). Consequently, the total $PM_{2.5}$
emission of February 2020 for China shows an obvious decrease compared to those of previous
years (Figure 3). The high temporal-resolution DEPE reveals the detailed changes of $PM_{2.5}$
emission with time (Figure 10). The $PM_{2.5}$ emission started to decrease right around the COVID
outbreak, and had been smaller than those of year 2019 till early March. The emissions at the
following months of 2020 are similar to those of 2019, due to the epidemic prevention and control
policies enforced by the China government. During February 2020, the DEPE shows significant
reductions at the north China plain and northeast of China where prominent $PM_{2.5}$ emission
occurred, while spotted $PM_{2.5}$ emission differences with small magnitudes showed at the other
regions (Figures 11a-b). Along with recovery from the COVID-19, the estimated $PM_{2.5}$ emission
rebounded in March (Figures 3a, 10, 11c-d), which is contributed to the national work resumption.
Thus, the DEPE is able to timely reflect the dynamic response of $PM_{2.5}$ emission to the COVID-

346 19.

To avoid fluctuations due to diurnal variations and monthly changes of $PM_{2.5}$ emission, 7-

day averaged $PM_{2.5}$ emission differences between year 2020 and 2019 are used to analyze the
dynamic impact of COVID-19 on $PM_{2.5}$ emission (Figure 12). Before the lockdown, there were
slight $PM_{2.5}$ emission differences over several provinces (Figures 12a-b). During the first week of
lockdown, $PM_{2.5}$ emission reduction larger than $5 \times 10^{-2}$ ($\mu g \cdot m^{-2} \cdot s^{-1}$) that is about 60%-70%
emission reduction, occurred at Hubei, Hunan, Guangdong, Anhui and Zhejiang provinces (Figure
12c). The $PM_{2.5}$ emission reduction extended to BTH and Shandong province during the second
week of lockdown (Figure 12d), and continuously spread to the three northeast provinces of China
during the third week of lockdown (Figure 12e). During the third week of lockdown, the increased
$PM_{2.5}$ emissions for BTH and SCR are possibly caused by the long national vocation of spring
holiday of year 2019 (Ji et al., 2018). The inhomogeneous spatial variations of $PM_{2.5}$ emissions
possibly relate with different traditions and policy enforcements for different provinces. The $PM_{2.5}$
emission reduction had been maintained over the central and northern China till early March when
the lockdown was lift (Figures 12f-i). Though it is hard to see continuous and consistent signal of
lockdown for the whole China, the timely DEPE can provide up-to-date guidance for quantifying
socioeconomic impacts from rare events with large emission changes such as the COVID-19.

Although there were significant reductions of $PM_{2.5}$ emissions over the central and northern

China in February 2020, a severe air pollution event occurred over the north China in early
February 2020. Previous studies have shown that the factors influencing the severe air pollution
event include the still intensive emissions from industrial, power and residential, unfavorable
meteorological condition, anomalously high humidity that promoted aerosol heterogeneous
chemistry, and secondary aerosol formation associated with increased atmosphere oxidants (Le et
al. 2020; Sulaymon et al. 2021; Li et al., 2021) .

## 6. Discussion

High temporal-resolution and dynamics-based estimations of $PM_{2.5}$ emission can be

objectively and optimally obtained by assimilating past and future observed surface $PM_{2.5}$
concentrations through flow-dependent error statistics. This advanced assimilation strategy can be
applied for emission estimates of other chemical species when corresponding observations are
available, and extend to observation types besides the surface concentrations, like the aerosol
optical depth (Liu et al., 2011; Choi et al., 2020). Moreover, current estimates of $PM_{2.5}$ emission
are lack of explicitly representations of primary and secondary $PM_{2.5}$, which could be resolved by
joint estimation of the source emission, primary and secondary $PM_{2.5}$ given the concentration
observations. Another deficiency of this top-down technique is that it cannot directly determine
dynamics-based $PM_{2.5}$ emissions for different sectors and contributions from different policies,
although the bottom-up technique has the potential to untangle the different contributions from
different policies and quantify the different impacts on different sectors. However, this top-down
technique can be integrated into the bottom-up technique to retain advantages of both methods.
One future work is to integrate the top-down technique with the bottom-up one, by which the
emission estimates for different sectors and polices could be quantified. The annual emission
estimate from the bottom-up technique can be further downscaled to hourly estimates by first
distributing the annual amount to each month through the monthly allocations estimated from the
top-down technique, and then assuming evenly daily distribution, finally applying the fractions of
diurnal variation estimated from the top-down technique. The information collected by the bottom-
up technique is retained, while the common drawback of coarse temporal resolution for the bottom-
up technique is remedied. The integrated bottom-up and top-down technique can improve
spatiotemporal representations of source emissions cross time scales and sectors, which is
beneficial for emission inventory, air quality forecast, regulation policy and emission trading
scheme.

**Acknowledgments**
This work is jointly sponsored by the National Key R&D Program of China through Grant
2017YFC1501603 and the National Natural Science Foundation of China through Grants
41922036 and 42275153. We are grateful to the High Performance Computing Center of Nanjing
University for doing the cycling ensemble assimilation experiments.
**Data availability**
The meteorological data used for meteorological initial conditions and boundary conditions
is available from the University Corporation for Atmospheric Research (UCAR) Research Data
Archive (https://rda.ucar.edu/datasets/ds083.3/). The assimilated meteorological observations are
available from the UCAR Research Data Archive (https://rda.ucar.edu/datasets/ds337.0/), and the
assimilated chemical observations are available from https://aqicn.org/map/china/cn/. The
prescribed time-invariant anthropogenic emissions are available from the Emission Database for
Global Atmospheric Research for Hemispheric Transport of Air Pollution (EDGAR-HTAP)
inventory (https://data.jrc.ec.europa.eu/dataset/jrc-edgar-htap_v2-2) and the Multi-resolution
Emission Inventory (MEIC; http://meicmodel.org/?page_id=560).
The    WRF-Chem    model    version    3.6.1    is    available    from
https://www2.mmm.ucar.edu/wrf/users/download/get_sources.html#WRF-Chem. The    NOAA
operational EnKF system is available from https://dtcenter.org/community-code/gridpoint-
statistical-interpolation-gsi.

**Competing interests**

The contact author has declared that none of the authors has any competing interests.

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

**Figures and Tables**
**Captions:**
**Figure 1.** Times series of hourly $PM_{2.5}$ concentration biases ($\mu g \cdot m^{-3}$). The ensemble mean priors
compared to the observed quantities for December of years 2016-2020 (gray and black), and the
mean biases of years 2016-2020 (blue).
**Figure 2.** Model domain and observation sites for cycling assimilation. Red and blue dots denote
the assimilated and unassimilated observational sites, respectively.
**Figure 3.** (a) Dynamics-based monthly $PM_{2.5}$ emission estimates ($Tg \cdot day^{-1}$) summed over
mainland China of each year from 2016 to 2020 (colored) and the estimated $PM_{2.5}$ emission from
MEIC (gray); (b) Ratio of $PM_{2.5}$ emission changes between two adjacent years from year 2016 to
2020 normalized by the $PM_{2.5}$ emission of year 2016; (c) Monthly fractions of dynamics-based
$PM_{2.5}$ emission estimates for years 2016-2020 (light blue), the five-year mean fractions of
dynamics-based monthly $PM_{2.5}$ emission estimates with bars denoting one standard deviation of
the five-year variations (dark blue), and the monthly fractions of estimated $PM_{2.5}$ emission from
MEIC (gray).
**Figure 4.** (a) Spatial distribution of dynamics-based $PM_{2.5}$ emission estimates ($\mu g \cdot m^{-2} \cdot s^{-1}$) for
year 2016, and compared to that of year 2016, spatial distributions of dynamics-based $PM_{2.5}$
emission changes of year (b) 2017, (c) 2018, (d) 2019 and (e) 2020.
**Figure 5.** (a) The differences of dynamics-based $PM_{2.5}$ emission estimates between years 2017-
2020 and 2016, and (b) the differences normalized by that of year 2016.
**Figure 6**. Spatial distributions of dynamics-based $PM_{2.5}$ emission changes in December compaered to November
in (a) 2017 and (b) 2018.**Figure 7.** Five-year mean diurnal variations of dynamics-based $PM_{2.5}$
emission fraction averaged over (a) mainland China, (b) megacities with urban population $\geq$ 5
million, and (c) non-megacities with urban population < 5 million.

**Figure 8.** Diurnal variations of dynamics-based PM$_{2.5}$ emission fractions for years 2016-2020 (light blue) and five-year mean fractions with bars denoting one standard deviation of the five-year variations (dark blue) are averaged over mainland China for (a) January, (b) April, (c) July, and (d) October.

**Figure 9.** Diurnal variations of PM$_{2.5}$ emission fraction for each month based on diurnal variation profiles from ES and EU (Wang et al. 2010).

**Figure 10.** Hourly (light red and blue) and daily (dark red and blue) dynamics-based PM$_{2.5}$ emission estimates (kg·h$^{-1}$) summed over mainland China from January to March of years 2019 and 2020.

**Figure 11.** Spatial distributions of dynamics-based PM$_{2.5}$ emission estimates (µg·m$^{-2}$·s$^{-1}$) on (b) February and (d) March of year 2019, and spatial distributions of dynamics-based PM$_{2.5}$ emission reduction of year 2020 compared to year 2019 for (c) February and (e) March.

**Figure 12.** Mean spatial distributions of PM$_{2.5}$ emission differences (µg·m$^{-2}$·s$^{-1}$) between year 2020 and 2019 for 9 weeks starting at 9 January 2020. Negative (positive) values indicate that PM$_{2.5}$ emission of year 2020 is smaller (larger) than that of year 2019. The numbers in (a) denote provinces as: 1 Heilongjiang, 2 Neimenggu, 3 Xinjiang, 4 Jilin, 5 Liaoning, 6 Gansu, 7 Hebei, 8 Beijing, 9 Shanxi, 10 Tianjin, 11 Shanxi, 12 Ningxia, 13 Qinghai, 14 Shandong, 15 Xizang, 16 Henan, 17 Jiangsu, 18 Anhui, 19 Sichuan, 20 Hubei, 21 Chongqing, 22 Shanghai, 23 Zhejiang, 24 Hunan, 25 Jiangxi, 26 Yunnan, 27 Guizhou, 28 Fujian, 29 Guangxi, 30 Guangdong, 31 Taiwan, 32 Hongkong, 33 Macao, 34 Hainan.

**Table 1.** Dynamics-based PM$_{2.5}$ emission estimates of year 2016 for each province whose value is larger than 0.01 µg·m$^{-2}$·s$^{-1}$ are shown in the second column. Ratios of PM$_{2.5}$ emission changes of years 2017-2020 compared to year 2016 are shown from the third to the sixth column, with negative (positive) values indicating decrease (increase) of PM$_{2.5}$ emission.

**Table 2.** Five-year mean diurnal fractions (%) of the dynamics-based PM$_{2.5}$ emission estimates over mainland China on local solar time (LST) for each month.


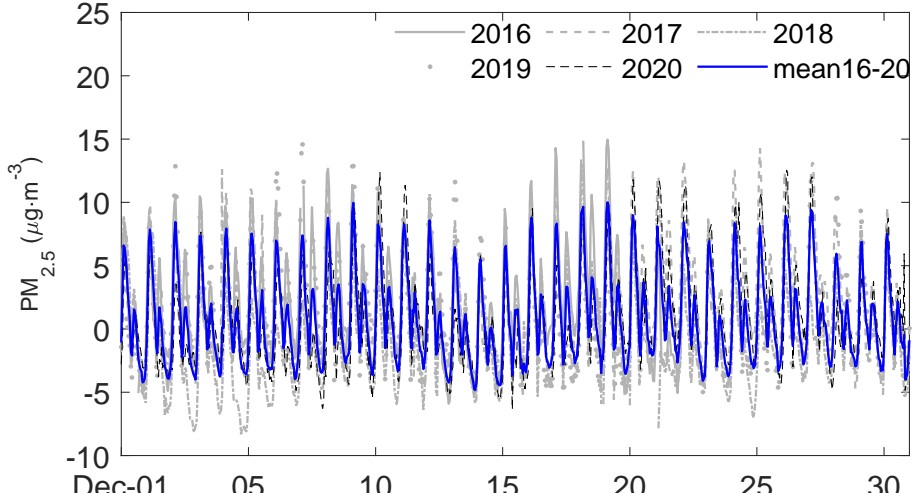

**Figure 1.** Times series of hourly PM$_{2.5}$ concentration biases (μg·m$^{-3}$). The ensemble mean priors compared to
the observed quantities for December of years 2016-2020 (gray and black), and the mean biases of years 2016-
2020 (blue).


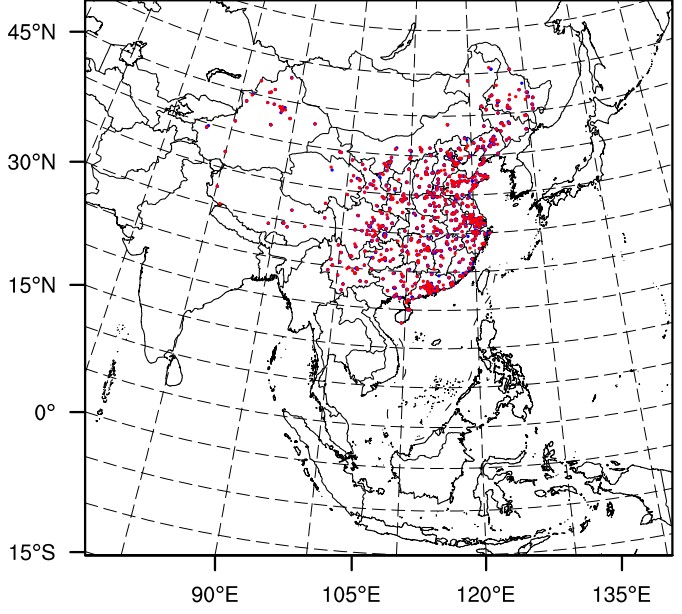

**Figure 2.** Model domain and observation sites for cycling assimilation. Red and blue dots denote the
assimilated and unassimilated observational sites, respectively.

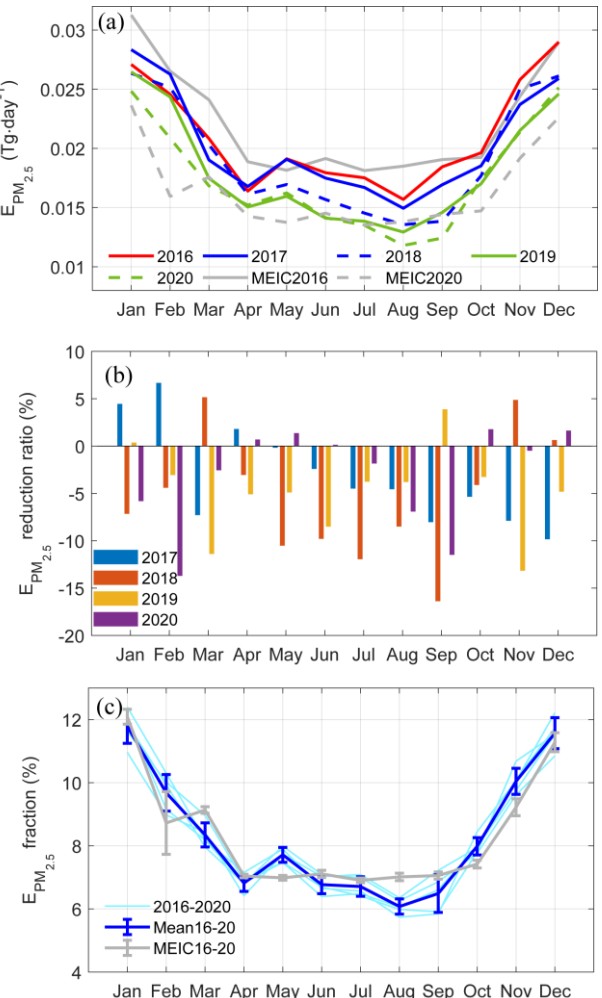

**Figure 3.** (a) Dynamics-based monthly PM$_{2.5}$ emission estimates (Tg·day$^{-1}$) summed over mainland China of
each year from 2016 to 2020 (colored) and the estimated PM$_{2.5}$ emission from MEIC (gray); (b) Ratio of PM$_{2.5}$
emission changes between two adjacent years from year 2016 to 2020 normalized by the PM$_{2.5}$ emission of
year 2016; (c) Monthly fractions of dynamics-based PM$_{2.5}$ emission estimates for years 2016-2020 (light blue),
the five-year mean fractions of dynamics-based monthly PM$_{2.5}$ emission estimates with bars denoting one
standard deviation of the five-year variations (dark blue), and the monthly fractions of estimated PM$_{2.5}$
emission from MEIC (gray).

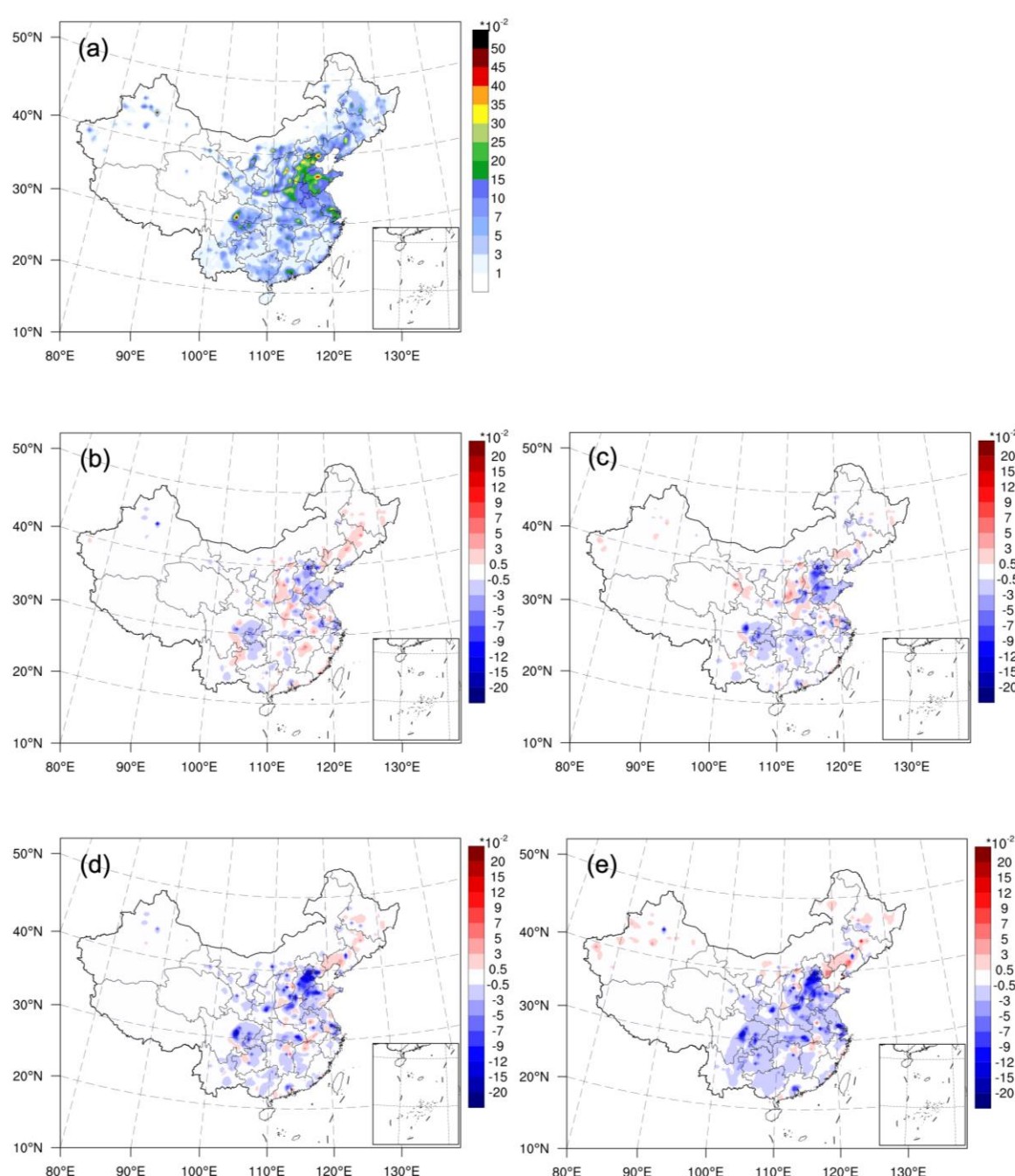

**Figure 4.** (a) Spatial distribution of dynamics-based PM$_{2.5}$ emission estimates ($\mu g \cdot m^{-2} \cdot s^{-1}$) for year 2016, and
compared to that of year 2016, spatial distributions of dynamics-based PM$_{2.5}$ emission changes of year (b)
2017, (c) 2018, (d) 2019 and (e) 2020.

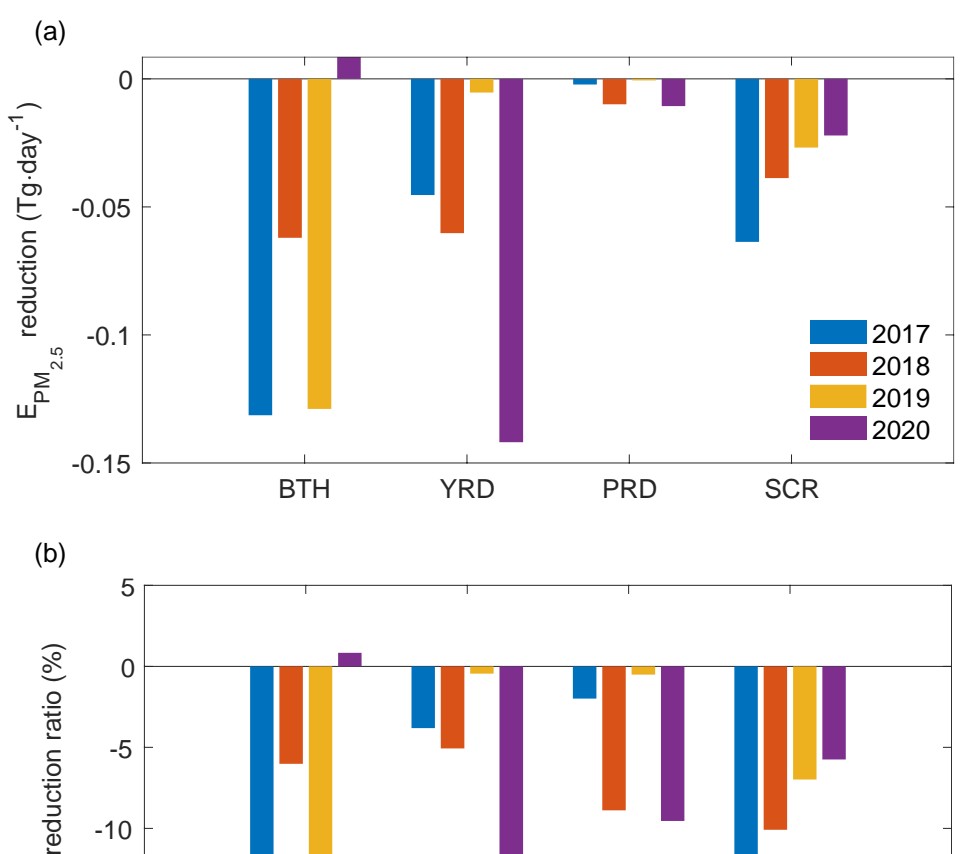

**Figure 5.** (a) The differences of dynamics-based PM$_{2.5}$ emission estimates between years 2017-2020 and 2016,
and (b) the differences normalized by that of year 2016.

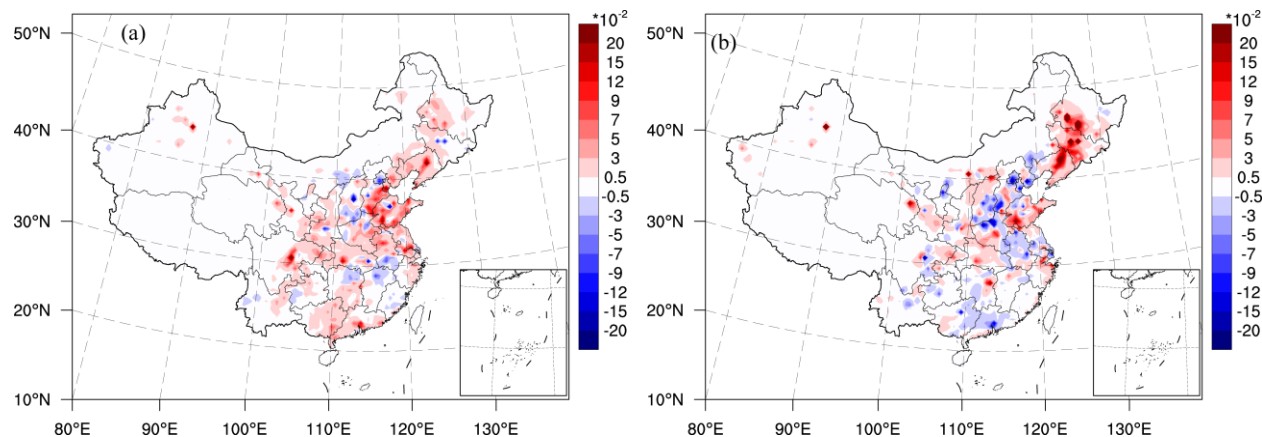

**Figure 6.** Spatial distributions of dynamics-based PM$_{2.5}$ emission changes in December compaered to
November in (a) 2017 and (b) 2018.

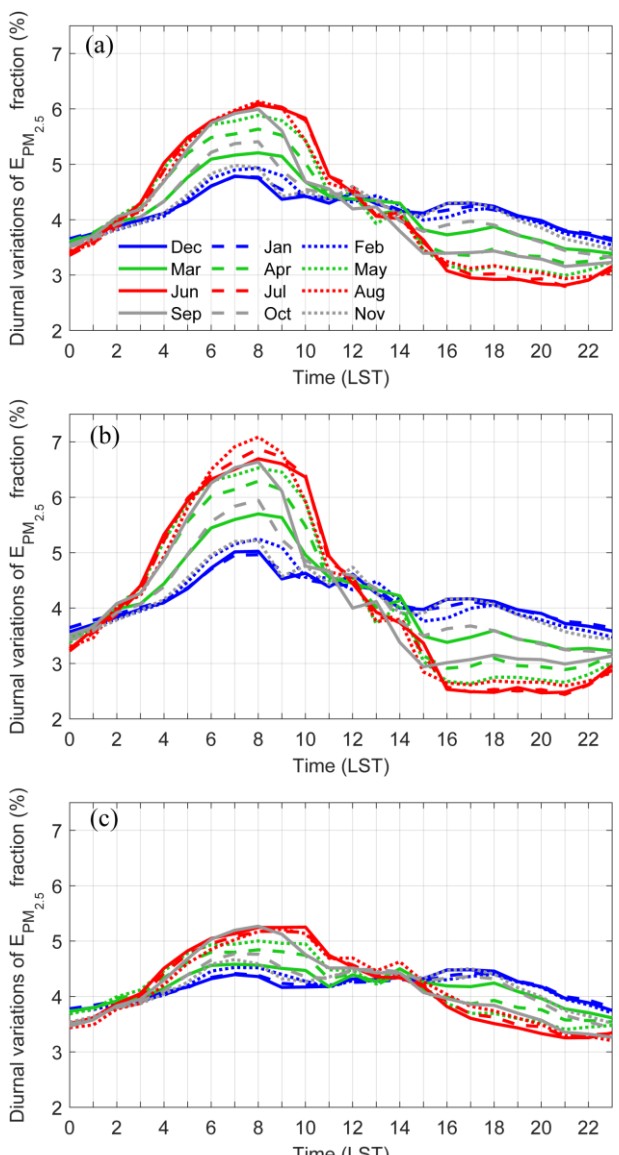


**Figure 7.** Five-year mean diurnal variations of dynamics-based $PM_{2.5}$ emission fraction averaged over (a) mainland
China, (b) megacities with urban population ≥ 5 million, and (c) non-megacities with urban population < 5 million.

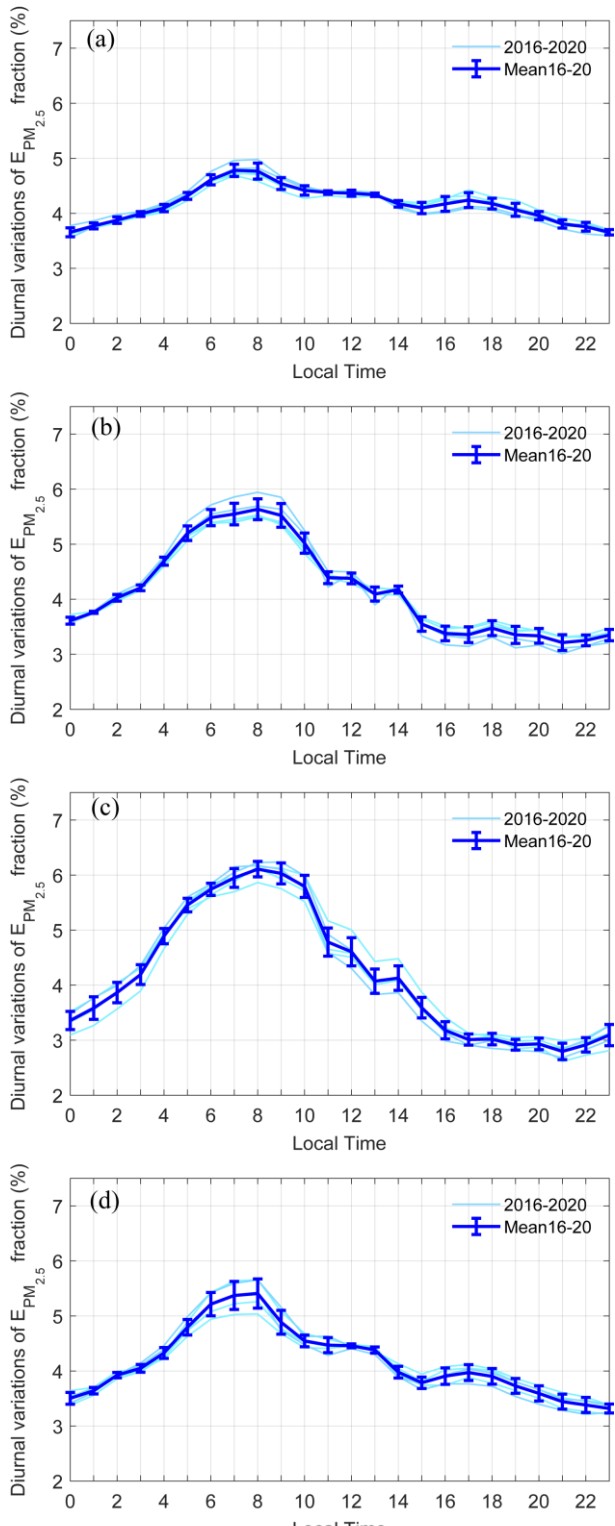

Figure 8. Diurnal variations of dynamics-based PM$_{2.5}$ emission fractions for years 2016-2020 (light blue) and five-year mean fractions with bars denoting one standard deviation of the five-year variations (dark blue) are averaged over mainland China for (a) January, (b) April, (c) July, and (d) October.


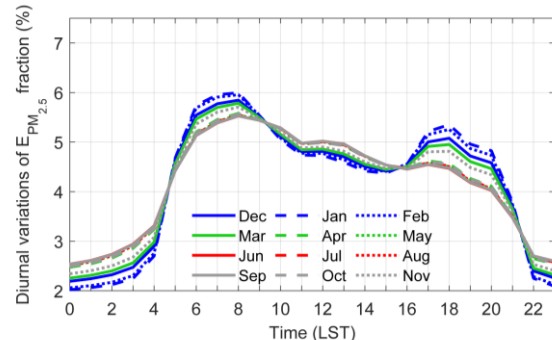

**Figure 9.** Diurnal variations of PM$_{2.5}$ emission fraction for each month based on diurnal
variation profiles from ES and EU (Wang et al. 2010).




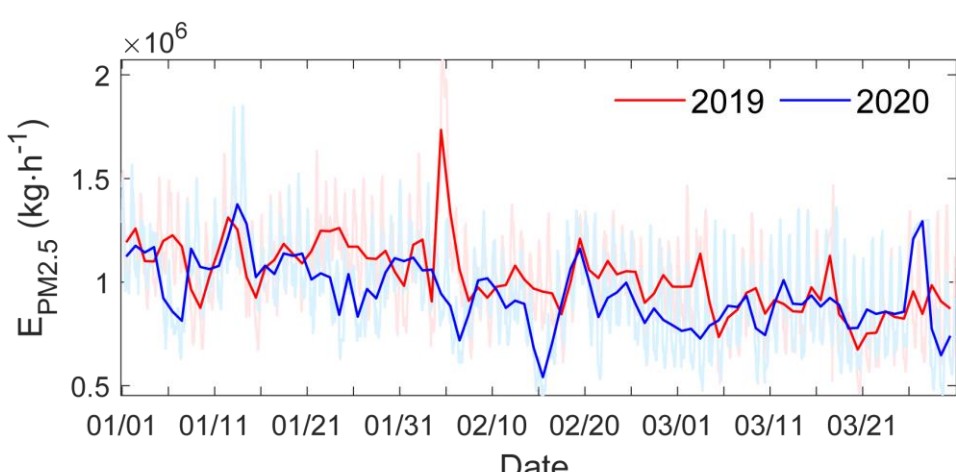


**Figure 10.** Hourly (light red and blue) and daily (dark red and blue) dynamics-based PM$_{2.5}$ emission estimates (kg·h$^{-}$
$^1$) summed over mainland China from January to March of years 2019 and 2020.


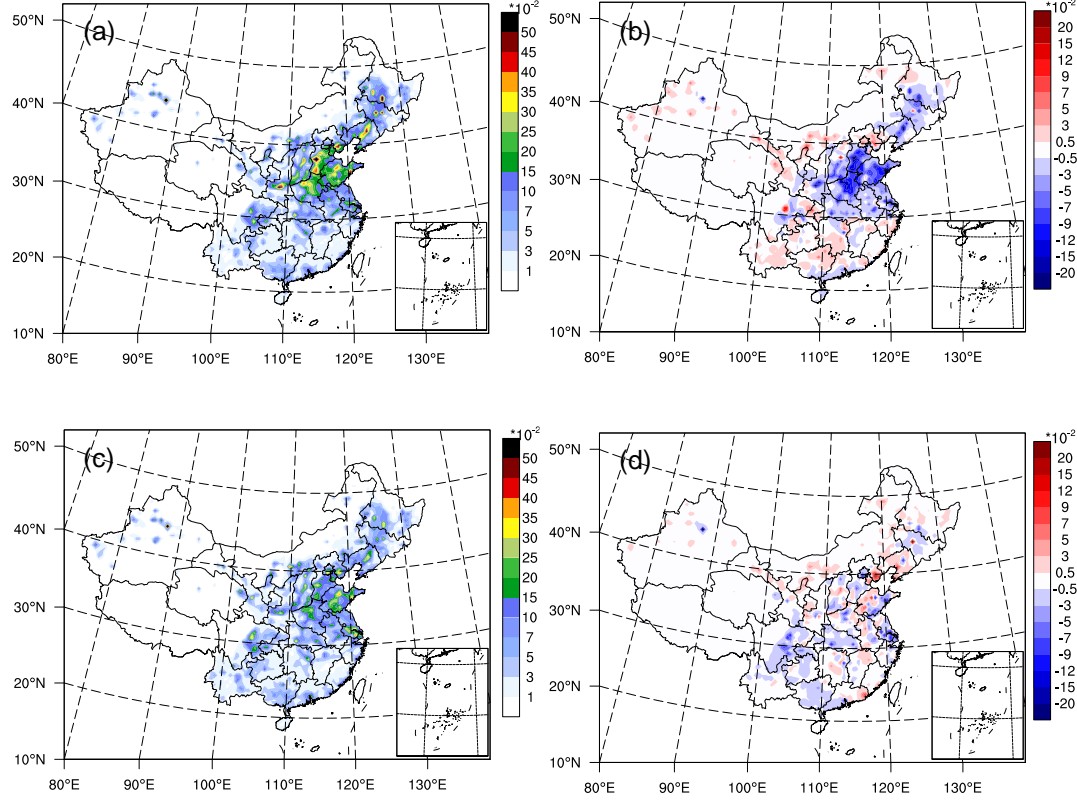

**Figure 11.** Spatial distributions of dynamics-based PM$_{2.5}$ emission estimates (μg·m$^{-2}$·s$^{-1}$) on (b) February and (d)
March of year 2019, and spatial distributions of dynamics-based PM$_{2.5}$ emission reduction of year 2020 compared to
year 2019 for (c) February and (e) March.

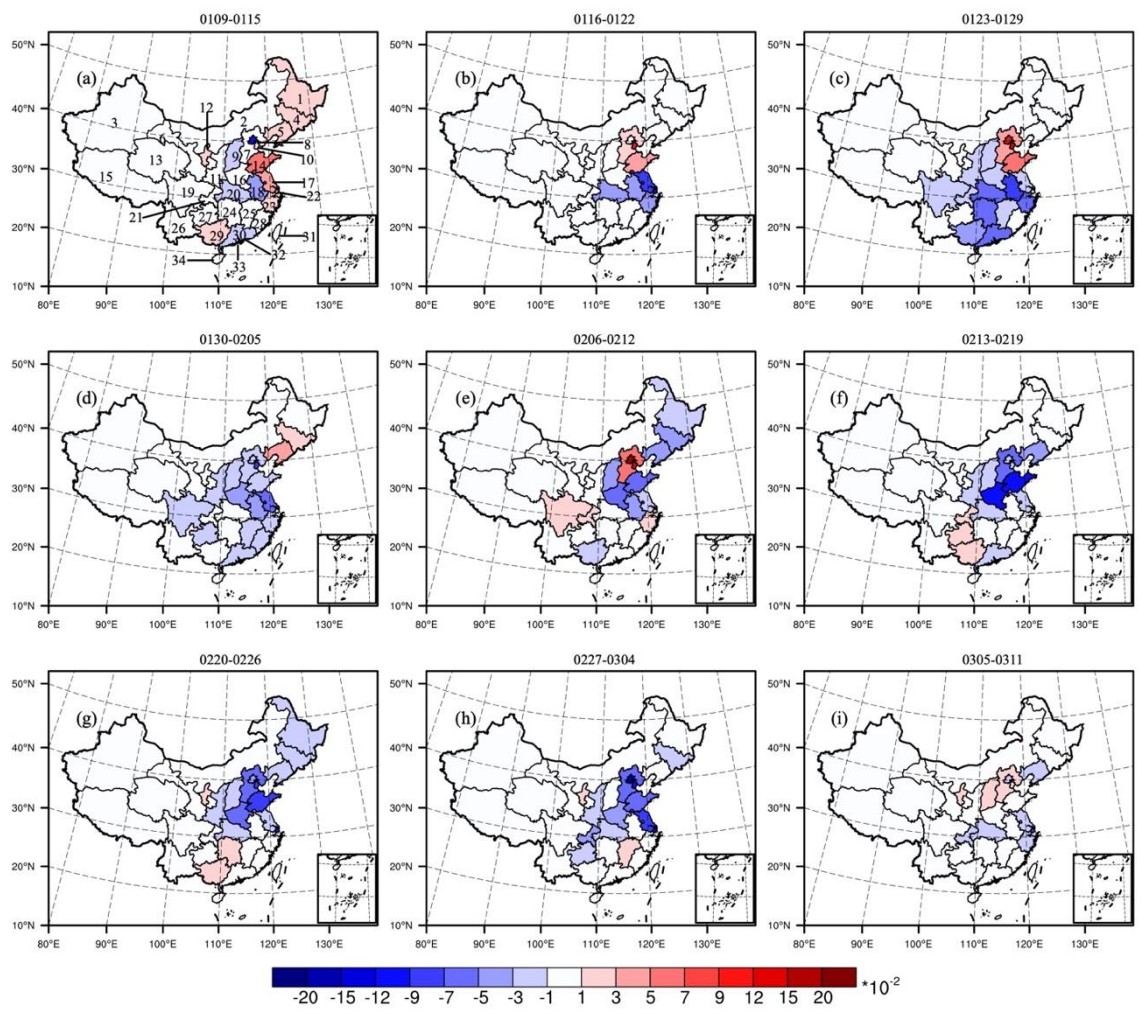

**Figure 12.** Mean spatial distributions of PM$_{2.5}$ emission differences (μg·m$^{-2}$·s$^{-1}$) between year
2020 and 2019 for 9 weeks starting at 9 January 2020. Negative (positive) values indicate that
PM$_{2.5}$ emission of year 2020 is smaller (larger) than that of year 2019. The numbers in (a) denote
provinces as: 1 Heilongjiang, 2 Neimenggu, 3 Xinjiang, 4 Jilin, 5 Liaoning, 6 Gansu, 7 Hebei, 8
Beijing, 9 Shanxi, 10 Tianjin, 11 Shanxi, 12 Ningxia, 13 Qinghai, 14 Shandong, 15 Xizang, 16
Henan, 17 Jiangsu, 18 Anhui, 19 Sichuan, 20 Hubei, 21 Chongqing, 22 Shanghai, 23 Zhejiang,
24 Hunan, 25 Jiangxi, 26 Yunnan, 27 Guizhou, 28 Fujian, 29 Guangxi, 30 Guangdong, 31
Taiwan, 32 Hongkong, 33 Macao, 34 Hainan.

Table 1. Dynamics-based $PM_{2.5}$ emission estimates of year 2016 for each province whose value is larger than $0.01\ \mu g \cdot m^{-2} \cdot s^{-1}$ are shown in the second column. Ratios of $PM_{2.5}$ emission changes of years 2017-2020 compared to year 2016 are shown from the third to the sixth column, with negative (positive) values indicating decrease (increase) of $PM_{2.5}$ emission.

| Province | $PM_{2.5}$ emission of year 2016 ($\mu g \cdot m^{-2} \cdot s^{-1}$) | Percentage of $PM_{2.5}$ emission change for year 2017 (%) | Percentage of $PM_{2.5}$ emission change for year 2018 (%) | Percentage of $PM_{2.5}$ emission change for year 2019 (%) | Percentage of $PM_{2.5}$ emission change for year 2020 (%) |
|---|---|---|---|---|---|
| Tianjin | 0.2083 | -14.07 | -22.99 | -38.70 | -26.98 |
| Shanghai | 0.2067 | -24.39 | -30.21 | -21.46 | -30.05 |
| Shandong | 0.1631 | -15.26 | -21.02 | -15.57 | -19.41 |
| Beijing | 0.1598 | -26.64 | -25.75 | -41.92 | -45.27 |
| Hebei | 0.1178 | -7.47 | -11.98 | -26.39 | -22.87 |
| Jiangsu | 0.1088 | -6.52 | -3.98 | -12.69 | -28.20 |
| Henan | 0.1064 | -1.41 | -3.68 | -12.15 | -24.91 |
| Shanxi | 0.0885 | 6.17 | 7.90 | -13.18 | -13.85 |
| Liaoning | 0.0742 | 6.32 | -2.58 | 3.22 | 11.42 |
| Anhui | 0.0687 | 1.92 | -5.63 | -6.23 | -21.57 |
| Hubei | 0.0574 | -5.87 | -17.69 | -19.76 | -36.48 |
| Zhejiang | 0.0557 | -3.62 | -9.32 | -9.99 | -18.05 |
| Chongqing | 0.0525 | -22.24 | -29.81 | -24.63 | -38.41 |
| Shanxi | 0.0498 | 0.62 | -1.97 | -18.05 | -17.85 |
| Guangdong | 0.0481 | 1.21 | -6.01 | -6.69 | -14.37 |
| Ningxia | 0.0481 | -8.17 | -5.93 | -24.46 | -12.95 |
| Hunan | 0.0417 | -6.40 | -19.35 | -9.91 | -20.62 |
| Guangxi | 0.0390 | -2.42 | -3.52 | -12.47 | -22.31 |
| Guizhou | 0.0365 | -4.01 | -15.82 | -21.74 | -46.41 |
| Jilin | 0.0360 | 12.30 | -3.22 | 7.37 | 4.76 |
| Jiangxi | 0.0353 | 13.22 | -9.67 | -7.19 | -11.91 |
| Sichuan | 0.0337 | -7.66 | -15.66 | -27.68 | -37.93 |
| Fujian | 0.0244 | 3.13 | -2.73 | -8.13 | -13.41 |
| Heilongjiang | 0.0231 | 7.30 | -0.21 | 3.14 | 3.91 |
| Yunnan | 0.0221 | -1.26 | -7.16 | -9.93 | -15.35 |
| Gansu | 0.0177 | -4.26 | 5.28 | -17.89 | -16.49 |
| Hainan | 0.0173 | 3.93 | -0.41 | -5.04 | -4.78 |
| Neimenggu | 0.0141 | -0.00 | -3.63 | -8.16 | 3.55 |

**Table 2.** Five-year mean diurnal fractions (%) of the dynamics-based PM$_{2.5}$ emission estimates
over mainland China on local solar time (LST) for each month.

| | Jan | Feb | Mar | Apr | May | Jun | Jul | Aug | Sep | Oct | Nov | Dec |
|---|---|---|---|---|---|---|---|---|---|---|---|---|
| 0 | 3.65 | 3.58 | 3.61 | 3.61 | 3.55 | 3.40 | 3.36 | 3.44 | 3.55 | 3.50 | 3.53 | 3.63 |
| 1 | 3.77 | 3.69 | 3.72 | 3.76 | 3.74 | 3.65 | 3.58 | 3.56 | 3.70 | 3.64 | 3.64 | 3.75 |
| 2 | 3.88 | 3.82 | 3.96 | 4.03 | 4.05 | 3.94 | 3.86 | 4.01 | 4.05 | 3.93 | 3.83 | 3.89 |
| 3 | 3.98 | 3.94 | 4.05 | 4.21 | 4.29 | 4.30 | 4.19 | 4.14 | 4.19 | 4.05 | 3.93 | 3.99 |
| 4 | 4.10 | 4.06 | 4.33 | 4.69 | 4.92 | 5.03 | 4.89 | 4.71 | 4.69 | 4.33 | 4.12 | 4.12 |
| 5 | 4.32 | 4.38 | 4.76 | 5.20 | 5.46 | 5.48 | 5.45 | 5.39 | 5.27 | 4.80 | 4.45 | 4.32 |
| 6 | 4.61 | 4.74 | 5.09 | 5.48 | 5.72 | 5.78 | 5.74 | 5.78 | 5.74 | 5.21 | 4.83 | 4.61 |
| 7 | 4.78 | 4.90 | 5.17 | 5.55 | 5.78 | 5.92 | 5.95 | 5.98 | 5.92 | 5.37 | 4.98 | 4.79 |
| 8 | 4.77 | 4.93 | 5.21 | 5.63 | 5.88 | 6.07 | 6.11 | 6.13 | 5.99 | 5.41 | 4.94 | 4.75 |
| 9 | 4.54 | 4.79 | 5.14 | 5.52 | 5.79 | 6.00 | 6.03 | 6.02 | 5.60 | 4.89 | 4.42 | 4.37 |
| 10 | 4.41 | 4.41 | 4.68 | 5.02 | 5.43 | 5.83 | 5.79 | 5.42 | 4.68 | 4.55 | 4.50 | 4.42 |
| 11 | 4.38 | 4.40 | 4.42 | 4.39 | 4.45 | 4.79 | 4.78 | 4.66 | 4.56 | 4.47 | 4.36 | 4.30 |
| 12 | 4.37 | 4.32 | 4.37 | 4.38 | 4.48 | 4.49 | 4.61 | 4.51 | 4.19 | 4.46 | 4.60 | 4.48 |
| 13 | 4.34 | 4.43 | 4.34 | 4.09 | 3.93 | 4.06 | 4.07 | 4.09 | 4.23 | 4.38 | 4.33 | 4.29 |
| 14 | 4.17 | 4.26 | 4.30 | 4.18 | 4.16 | 4.02 | 4.13 | 4.10 | 3.79 | 3.98 | 4.10 | 4.15 |
| 15 | 4.10 | 3.99 | 3.82 | 3.55 | 3.46 | 3.63 | 3.59 | 3.45 | 3.39 | 3.79 | 4.07 | 4.12 |
| 16 | 4.17 | 4.05 | 3.73 | 3.38 | 3.17 | 3.08 | 3.18 | 3.24 | 3.40 | 3.92 | 4.30 | 4.29 |
| 17 | 4.24 | 4.17 | 3.79 | 3.36 | 3.08 | 2.95 | 3.01 | 3.12 | 3.41 | 3.98 | 4.31 | 4.30 |
| 18 | 4.18 | 4.21 | 3.87 | 3.48 | 3.16 | 2.92 | 3.03 | 3.17 | 3.44 | 3.91 | 4.21 | 4.24 |
| 19 | 4.06 | 4.04 | 3.72 | 3.35 | 3.12 | 2.92 | 2.93 | 3.08 | 3.34 | 3.73 | 3.99 | 4.07 |
| 20 | 3.96 | 3.93 | 3.62 | 3.34 | 3.07 | 2.84 | 2.93 | 3.04 | 3.29 | 3.59 | 3.85 | 3.98 |
| 21 | 3.81 | 3.75 | 3.47 | 3.21 | 2.99 | 2.83 | 2.80 | 2.93 | 3.16 | 3.44 | 3.65 | 3.79 |
| 22 | 3.76 | 3.66 | 3.44 | 3.25 | 3.09 | 2.91 | 2.92 | 2.97 | 3.19 | 3.38 | 3.56 | 3.73 |
| 23 | 3.65 | 3.55 | 3.39 | 3.34 | 3.23 | 3.16 | 3.09 | 3.04 | 3.23 | 3.32 | 3.47 | 3.62 |



**Appendix: Effects of meteorology**
An observing system simulation experiment (OSSE) is performed to investigate the effects of time-
varying boundary layer. A nature run is first conducted from 0000 UTC 25 December 2015 to
0000 UTC 2 February 2016, forced by the time-invariant source emissions PR2010 (the true
emission). Synthetic observations of the six conventional air pollutant concentrations (i.e., $PM_{10}$,
$PM_{2.5}$, $SO_2$, $NO_2$, $O_3$, and CO) are generated from the natural run. Hourly synthetic observations
are created from 0000 UTC 29 December 2015 to 0006 UTC 1 February 2016, by interpolating
the gridded true surface concentrations to the chemical observation locations with additive random
errors of $N(0,R)$. $R$ is the observation error variance, which is calculated by the formula in Elbern
et al. (2007). Outputs from the first four days of the natural run are excluded to avoid the transient
effect. Then the prior emissions are generated by $F^{pr} = (1.8 + \delta(x, y, z, t))F^{tr}$, where $F^{tr}$ is the true
emission, $\delta$ is a random number sampled from the normal distribution $N(0,1)$ (Peng et al. 2015).
Ensemble data assimilation experiments are conducted from 0000 UTC 29 December to 0006 UTC
1 February 2016. Outputs from the first two days of the OSSE are excluded due to the spin-up.
The magnitude of posterior $PM_{2.5}$ emission is closer to the true emission than the prior. Figure S1
presents the monthly mean diurnal variations of $PM_{2.5}$ emission fraction from the OSSE. It shows
that a little larger estimated $PM_{2.5}$ emission fractions occurred in the morning and smaller
estimated $PM_{2.5}$ emission fractions occurred in the afternoon, comparing to the time-invariant true
emission. But the diurnal variations of $PM_{2.5}$ emission fractions caused by the boundary layer are
not as strong as that caused by the emission itself (Figure 7). The reason may be that we have
hourly assimilated observations to simultaneously update the chemical concentrations and source
emissions. Therefore, the impacts of time-varying boundary layer on the posterior $PM_{2.5}$ emissions
are limited.

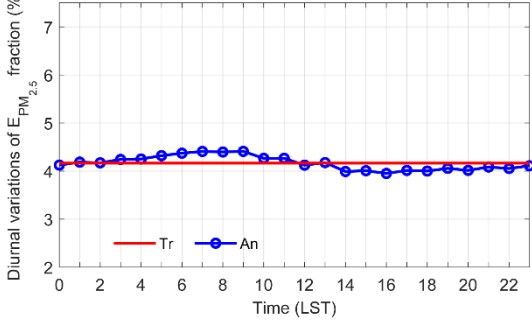


**Figure S1.** Diurnal variations of PM$_{2.5}$ emission fraction for the Observing System Simulation

Experiment.