# Peer review of "Dynamics-based estimates of decline trend with fine temporal variations in China's"

_EGUsphere, 2023_

## Author Comment (AC1)

**Response to Reviewer #1**

Thank the reviewer for the insightful and detailed comments and suggestions, which helped to significantly improve the manuscript.

The reviewer's comments are shown in *blue italics* with the author responses in black.

*This study assimilates hourly air quality observations to provide detailed primary PM2.5 emissions inventory across China. Some insights about inter-annual, monthly and diurnal variation of the derived emissions, as well as impacts from COVID-19 lockdown are discussed. While the scope of this study is definitely within the scope of ACP, I have a serious reservation of publishing the manuscript at the present form. The main issue is about inadequate/insufficient discussion of the results and missing information of fundamentals which I will outline later. I am happy to re-review the revised manuscript, and will support the publication of this paper, if the following concerns can be adequately addressed.*

*Major points:*

*1) Assimilation approach regarding secondary PM2.5: it is hard for me to understand (Page 6-7) how "the impact of the secondary PM2.5 is ignored" during the assimilation. Secondary aerosol makes significant contributions (>50%) to observed PM2.5. This fact almost applies to all the hourly PM2.5 observations. Observations that are very remote and during less photochemically reactive periods may be less constituted by secondary aerosols, but I think these records make minor contributions to the air quality records this paper used which distribute mostly over eastern China. WRF-Chem simulation of PM2.5 includes important secondary species, which is fundamental to successfully capture PM2.5 spatiotemporal variations in China. So please further clarify:*

*What emission species are exactly optimized during the assimilation? Do you only constrain PM2.5 emissions and let PM2.5 precursors (SO2, NH3, NOx, VOC) to stay the same as the a priori? If so, how uncertain are the constrained PM2.5 emissions, if the a priori precursor emissions are incorrect and they make significant contributions to the observed PM2.5?*

*Lines 142 suggests that "hourly observed ... PM10, PM2.5, SO2, NO2, O3, and CO" are assimilated. So maybe these concentrations are used to also constrain the PM2.5*

*precursor emissions at the same time? If so, you should also briefly present the results of optimized precursor emissions, and how they affect the constraints on PM2.5 emissions.*

*Overall, I do not understand how successful constraints on PM2.5 emissions can be achieved with "ignoring secondary aerosol".*

Thank the reviewer for the valuable comments and suggestions. We agree with the reviewer that secondary aerosol is important to observed $PM_{2.5}$ and $PM_{2.5}$ precursors ($SO_2$, $NH_3$, $NO_x$, VOC) play crucial roles in the formation of secondary aerosol. The WRF-Chem model has good performances for simulating the spatiotemporal variations of $PM_{2.5}$ over China, due to the model advances, especially with the improved description of the secondary formation mechanisms of aerosols. We did not clearly describe the assimilation process related with the secondary formation. As the reviewer commented, the aerosol secondary formation is captured by the WRF-Chem model, while the $PM_{2.5}$, $O_3$, $NH_3$, and $PM_{2.5}$ precursors that have observations ($SO_2$ and NO), are updated by the observed quantities, respectively, but the VOC that are also $PM_{2.5}$ precursors are not updated due to the lack of direct and limited observations.

The detailed updated chemical variables and emission species follow Peng et al. (2018). Six conventional air pollutant observations (i.e., $PM_{10}$, $PM_{2.5}$, $SO_2$, $NO_2$, $O_3$, and CO) are obtained from the Ministry of Ecology and Environment of China. The concentrations and emissions related to the six observations are constrained by the observed quantities. $PM_{2.5}$ observations are used to constrain the mass concentrations of $P_{2.5}$ (the fine unspeciated aerosol contributions), S (sulfate), $OC_1$ (hydrophobic organic carbon), $OC_2$ (hydrophilic organic carbon), $BC_1$ (hydrophobic black carbon), $BC_2$ (hydrophilic black carbon), $D_1$ (dusts with effective radii of 0.5 μm), $D_2$ (dusts with effective radii of 1.4 μm), $S_1$ (sea salts with effective radii of 0.3 μm) and $S_2$(sea salts with effective radii of 1.0 μm). Besides, the emissions of the unspeciated primary sources of $PM_{2.5}$, sulfate, nitrate and $NH_3$ are also updated. $PM_{10-2.5}$ observations (the differences between the $PM_{10}$ observations and the $PM_{2.5}$ observations) are used to constrain the mass concentrations of $P_{10}$ (the coarse-mode unspeciated aerosol contributions), $D_3$ (dusts with effective radii of 2.4 μm), $D_4$ (dusts with effective radii of 4.5 μm), $D_5$, $S_3$(sea salts with effective radii of 3.25 μm), $S_4$ and the emission of $PM_{10}$. $SO_2$ observations are used to constrain the $SO_2$ concentrations and emissions.

NO$_2$ observations are used to update the NO and NO$_2$ concentrations and NO emissions. CO observations are used to constrain the CO concentrations and emissions, while O$_3$ observations are used to constrain the O$_3$ concentrations.

To clarify the details of the assimilation process and the treatment for the secondary formation, the text in Lines 171-177, Page 7, is added. The details of the updated chemical and emission species follow Peng et al. (2018), which is not include in the text.

"Since the secondary formation process can be captured by the WRF-Chem model, the impact of the secondary PM$_{2.5}$ is indirectly considered. The detailed updated state variables with the according observations follow Peng et al. (2018). The concentrations and emissions of PM$_{2.5}$, NH$_3$, and PM$_{2.5}$ precursors that have observations (SO$_2$ and NO), are updated by the observed quantities, respectively, but the VOC that are also PM$_{2.5}$ precursors are not updated due to the lack of direct and limited observations."

*2) A missing piece of information is showing the improvement in model simulation/prediction after the assimilation? How is the agreement of simulated PM2.5 vs. observations improved after the assimilation? If you also constrained precursor emissions, comparison vs. observed air quality species other than PM2.5 should also be provided.*

*Another suggestion is to compare your results vs. the updated MEIC that has extended to more recent years (not just 2016). This discussion is especially necessary considering that MEIC contains detailed bottom-up information. Differences of the derived inter-annual and inter-month variations of emissions vs. MEIC will be indicative where and when MEIC might be unrepresentative and why.*

Thank the reviewer for the insightful suggestions. In this study, the hourly forecasts of PM$_{2.5}$ concentration from the cycling assimilation experiment matched the independent observed quantities well, as shown by Figure 1. This indicates that PM$_{2.5}$ simulations are improved by the assimilation. Moreover, our previous work demonstrated that simulations forced by the posterior emissions could produce smaller biases and errors for PM$_{2.5}$, SO$_2$, and NO$_2$ than those with a priori emissions, when independent verifications against the observed quantities (Peng et al., 2020). This discussion is added in the text in Lines 139-141, Page 6.

The manuscript is updated by extending the comparison to MEIC from 2016 to 2020. For years 2016-2020, the $PM_{2.5}$ emissions from MEIC are 8.10, 7.60, 6.70, 6.38 and 6.04 Tg, respectively. The $PM_{2.5}$ emissions from DEPE are 8.17, 7.91, 7.53, 7.13 and 6.89 Tg, respectively. However, MEIC focuses on the contributions of the anthropogenic activities, but does not consider the contributions of biomass burning emissions. Meanwhile, the DEPE has the posterior $PM_{2.5}$ emissions updated by the EnKS with both contributions from the anthropogenic and biomass burning emissions. The mean annual $PM_{2.5}$ emission from biomass burning in China (2003~2017) is 0.51 Tg (Yin et al., 2019). Thus to fairly compare with the MEIC, we subtract the contributions of biomass burning emissions from the dynamics-based estimates of $PM_{2.5}$ emissions, which gives posterior $PM_{2.5}$ emissions of 7.66, 7.40, 7.02, 6.62 and 6.38 Tg for years 2016-2020, respectively. The text and figures are all updated by the extended comparisons with the MEIC from years 2016-2020.

Yin, L., Du, P., Zhang, M., Liu, M., Xu, T., and Song, Y.: Estimation of emissions from biomass burning in China (2003–2017) based on MODIS fire radiative energy data, Biogeosciences, 16, 1629–1640, https://doi.org/10.5194/bg-16-1629-2019, 2019.

*3) Section 5: This section attributes the difference of 2020 emissions vs. the previous years to the COVID-19 lockdown. However, the 2020 emission vs. 2019 is not entirely stronger than the difference between other neighboring years (e.g., Figure 3 and Table 1). So how much of the 2020-2019 emission difference can also be contributed by continuous environmental policies (as discussed in Section 3)? Overall, Figure 11 does not provide continuous signal of lockdown either, as some provinces show temporary increases at certain phases. The authors discuss New Year firework. But how can they only occur in certain provinces (and not occurring during the first several days of New Year)? Overall, the attribution of 2020-2019 emission difference to COVID-19 lockdown and the relevant discussions about temporal changes of these differences are weak.*

 Thank the reviewer for the valuable comment.

We agree with the reviewer that the 2020 emission vs. 2019 is not entirely stronger than the differences between other neighboring years. As shown by Figure 3, the lockdown impact is mainly shown in February 2020. The emissions at the following months of February 2020 are similar to those of 2019, due to the epidemic prevention and control

policies enforced by the China government. This discussion is added in the text in Lines 338-340, Page 12.

The current EnKS, as a top-down technique, is unable to provide dynamics-based $PM_{2.5}$ emission estimates for different sectors and contributions from different policies, although the bottom-up technique has the potential to untangle the different contributions from different policies and quantify the different impacts on different sectors. One future work is to integrate the top-down technique with the bottom-up one, by which the emission estimates for different sectors and polices could be quantified. This discussion is added in the section of discussion in Lines 380-385, Page 14.

We agree with the reviewer that it is hard to see continuous and consistent signal of lockdown for the whole China, as shown by the original Figure 11 (new Figure 12). It is very likely that the inhomogeneous spatial variations of $PM_{2.5}$ emissions possibly relate with different traditions and policy enforcements for different provinces. As the third reviewer pointed out, the increased $PM_{2.5}$ emissions for BTH and SCR are not resulted from the firework, but possibly led by the long national vocation of spring holiday of year 2019 (Ji et al., 2018). This discussion is added in the text in Lines 356-358, 360-361, Page 13.

*Specific comments:*

*1) Abstract: key quantitative results should be presented. The current form of abstract is too qualitative and less informative. Line 24-27 reads redundant and irrelevant, and is suggested to be replaced with a more concise sentence stating the significance of these results.*

Thank the reviewer for the valuable comment. The abstract is modified to emphasize the findings and significance of the study. The updated abstract is:

"Timely, continuous, and dynamics-based estimates of $PM_{2.5}$ emissions with a high temporal resolution can be objectively and optimally obtained by assimilating observed surface $PM_{2.5}$ concentrations using flow-dependent error statistics. The annual dynamics-based estimates of $PM_{2.5}$ emission averaged over mainland China for years 2016-2020 without biomass burning emissions are 7.66, 7.40, 7.02, 6.62 and 6.38 Tg, respectively, which are very closed to the values of MEIC. Annual $PM_{2.5}$ emissions in China have consistently decreased of approximately 3% to 5% from 2017 to 2020.

Significant $PM_{2.5}$ emission reductions occurred frequently in regions with large $PM_{2.5}$ emissions. COVID-19 could cause a significant reduction of $PM_{2.5}$ emissions in the north China plain and northeast of China in 2020. The magnitudes of $PM_{2.5}$ emissions were greater in the winter than in the summer. $PM_{2.5}$ emissions show an obvious diurnal variation that varies significantly with the season and urban population. Compared to the diurnal variations of $PM_{2.5}$ emission fractions estimated based on diurnal variation profiles from US and EU, the estimated $PM_{2.5}$ emission fractions are 1.25% larger during the evening, the morning peak is 0.57% smaller in winter and 1.05% larger in summer, and the evening peak is 0.83% smaller. Improved representations of $PM_{2.5}$ emissions across time scales can benefit emission inventory, regulation policy and emission trading schemes, particularly for especially for high temporal resolution air quality forecasting and policy response to severe haze pollutions or rare human events with significant socioeconomic impacts."

*2) Why are observations before 2016 not used in the assimilation?*

The experimental period is from 2016 to 2020. We did not start the experiment earlier than 2016, due to the lack of concentration observations.

*3) Line 154: what is the spatial autocorrelation before and after the selection of stations?*

The spatial autocorrelation of the thinning of observations is close to the original observations. We have added this sentence in Line 155-156, Page 6.

*4) Line 196: What are the "weather effects" referring to?*

Thank the reviewer for pointing out this sentence. We realized that $PM_{2.5}$ emission is not related to the weather conditions, liking $PM_{2.5}$ concentrations. So we have moved this sentence in Line 220, Page 8.

*5) Figure 3b: the winter seasons show a sharp change from increases in 2017 to decreases in 2018. Is it related to the coal ban for residential heating since the 2017-2018 winter?*

The winter seasons demonstrate a sharp change from increases in 2017 to decreases in 2018. It is related to the coal ban for residential heating since the 2017-2018 winter. In 2017, spatial distributions of the changes of $PM_{2.5}$ emissions in December compared to November show obvious increases in most China. However, the changes in 2018 show

significant decreases in areas of Beijing, Tianjin, Hebei, Shanxi, Henan and Anhui provinces. This discussion is added in the text at 236-242, Page 9:

"The monthly DEPE also demonstrates the effectiveness of strict implementations of emission reduction policies in China, such as the coal ban for residential heating since the 2017-2018 winter. There was a sharp change of PM$_{2.5}$ emission, from increase in 2017 to decrease in 2018. As shown by Figure 6, spatial distributions of the changes of PM$_{2.5}$ emissions in December compared to November in 2017 show obvious increases in most China. However, the changes in 2018 show significant decreases in areas of Beijing, Tianjin, Hebei, Shanxi, Henan and Anhui provinces due to the implementation of the coal ban."

*6) Line 215-216: As I understand, the centralized heating system in North China has a fixed date of turning-on and turning-off during each heating season. So a sudden drop of emissions from March to April looks reasonable to me. Do you suggest that the turning-off date is variable in different places to smooth-out the differences, or residential heating does not contribute that much to the total emissions variations between these two months?*

We agree with the reviewer that the centralized heating system in North China has a fixed date of turning-on and turning-off during each heating season. Therefore, a sudden raise of emissions from October to November and a sudden drop of emissions from March to April is reasonable. But the turning-on and turning-off date are variable in different places, which help to smooth-out the differences. This discussion is added in the text at 248-252, Page 9:

"The centralized heating system in North China has a fixed date of turning-on and turning-off during each heating season. Therefore, a sudden raise of emissions from October to November and a sudden drop of emissions from March to April are shown. But the turning-on and turning-off date are variable in different regions, which imposes a smoothing impact on the emissions."

*7) Section 4: some recent bottom-up developments have more details about diurnal emission variations (e.g., Du et al., 10.5194/acp-20-2839-2020, 2020, Figure 1). Discussion about comparison of your results vs. these recent diurnal profiles can be insightful.*

Thank the reviewer for the thoughtful suggestion. As shown in Figure R1, the diurnal variation profiles for the power, industry, residential in Du et al. (2020) are the same as in Wang et al. (2010). However, the diurnal variation for transportation in Wang et al. (2010) are divided into four subcategories: light duty vehicles emissions in weekday, light duty vehicles emissions in weekend, heavy duty vehicles in weekday and heavy duty vehicles in weekend, which is more carefully than those in Du et al. (2020). So we did not compare the diurnal variation in Du et al. (2020), but we added this reference in Line 316, Page 12.

[Figure]

**Figure R1**. Diurnal variations of emissions from the individual sectors from (a) Wang et al., 2010; (b) Du et al., 2020.

*8) Line 310: missing words here.*

Thank the reviewer for pointing out the mistake. We have revised the text in Line 362, Page 13.

---

## Author Comment (AC2)

**Response to Reviewer #2**

We thank reviewer for his thoughtful comments and suggestions that have helped to significantly improve this manuscript.

The reviewer's comments are shown in *blue italics* with the author responses in black.

*Summary*

*This manuscript proposes an ensemble Kalman smoother to constrain the PM 2.5 emissions by incorporating the information of PM 2.5 observations. Results based on 5-year cycling assimilation provide quantitively estimates for annual and monthly variations of the PM 2.5 emission. By assimilating the observations with the ensemble Kalman smoother, the influences of COVID are clearly displayed. Moreover, diurnal variations of the PM 2.5 emission for each month are provided, which can be a valuable contribution to the PM 2.5 forecast. The manuscript proposed an advanced data assimilation method to update the PM 2.5 emissions by both present and future PM 2.5 observations. Overall it is well written and presented. It could be very beneficial to the community of chemistry data assimilation. I have several minor comments below.*

*1. An EnKS is proposed to update the emission along with the concentration. Are both the emission and the concentration updated by future observations?*

Thank the reviewer for the valuable comments.  The emissions are updated by current and future observations. But the concentrations are updated by current observations.

This discussion is added in the text in Lines 93-95, Page 4.

*2. The lagged length for EnKS is an important factor because it determines how many future observations are applied to constrain the current state. The lagged length K is set to 6 in this study. How this parameter is determined?*

Thank the reviewer for the valuable comments. The larger the K value, the more future observations are assimilated to constrain the current emission estimate. But the sample estimated temporal correlations could be contaminated by sampling errors and model errors, especially with increased lagged times. Thus, there is a tradeoff between the amount of future observations and accuracy of sample estimated temporal correlations. The choice of K (=6) is determined by sensitivity experiments. This discussion is added in the text in Lines 190-194, Pages 7-8

*3. It is interesting to see the quick influences of COVID on PM2.5 (Figure 11). Can such a DA system be practical for real-time operations?*

Thank the reviewer for the valuable comments. This DA system can be practical for real time operations.

---

## Author Comment (AC3)

**Response to Reviewer #3**

We thank reviewer for his thoughtful comments and suggestions that have helped to significantly improve this manuscript.

The reviewer's comments are shown in *blue italics* with the author responses in black.

*This study develops and presents top-down estimates of high temporal (up to hourly) PM2.5 emissions using an ENKS. The goal of this study is adequate for a publication in this journal, and I expect this research would inspire other researchers and would lead to further advances in top-down estimates of pollutant emissions. However, there are some parts that can be misleading or need to be clarified. Also, I agree with two other reviewers who raised important issues, which are as follows.*

*1. It is not clear how secondary PM2.5 is ignored. Did you just assume that the increments or differences resulting from PM2.5 assimilation are all attributed to PM2.5 emissions? Or, the formation of secondary PM2.5 is ignored in the WRF-Chem modeling? I agree with the first reviewer who emphasized the importance of secondary PM2.5 formation. The authors should demonstrate how the ignorance of secondary PM2.5 can be justified and what potential errors are.*

Thank the reviewer for the valuable comment. We agree with both Reviewer #1 and Reviewer #3 that secondary aerosol is important to observed $PM_{2.5}$ and $PM_{2.5}$ precursors ($SO_2$, $NH_3$, $NO_x$, VOC) play crucial roles in the formation of secondary aerosol. We did not clearly describe the assimilation process related with the secondary formation. As Reviewer #1 commented, the aerosol secondary formation is captured by the WRF-Chem model, while the $PM_{2.5}$, $O_3$, $NH_3$, and $PM_{2.5}$ precursors that have observations ($SO_2$ and NO), are updated by the observed quantities, respectively, but the VOC that are also $PM_{2.5}$ precursors are not updated due to the lack of direct and limited observations.

The detailed updated chemical variables and emission species follow Peng et al. (2018). Six conventional air pollutant observations (i.e., $PM_{10}$, $PM_{2.5}$, $SO_2$, $NO_2$, $O_3$, and CO) are obtained from the Ministry of Ecology and Environment of China. The concentrations and emissions related to the six observations are constrained by the observed quantities. $PM_{2.5}$ observations are used to constrain the mass concentrations of $P_{2.5}$ (the fine unspeciated aerosol contributions), S (sulfate), $OC_1$ (hydrophobic

organic carbon), $OC_2$ (hydrophilic organic carbon), $BC_1$ (hydrophobic black carbon), $BC_2$ (hydrophilic black carbon), $D_1$ (dusts with effective radii of 0.5 μm), $D_2$ (dusts with effective radii of 1.4 μm), $S_1$ (sea salts with effective radii of 0.3 μm) and $S_2$(sea salts with effective radii of 1.0 μm). Besides, the emissions of the unspeciated primary sources of $PM_{2.5}$, sulfate, nitrate and $NH_3$ are also updated. $PM_{10-2.5}$ observations (the differences between the $PM_{10}$ observations and the $PM_{2.5}$ observations) are used to constrain the mass concentrations of $P_{10}$ (the coarse-mode unspeciated aerosol contributions), $D_3$ (dusts with effective radii of 2.4 μm), $D_4$ (dusts with effective radii of 4.5 μm), $D_5$, $S_3$(sea salts with effective radii of 3.25 μm), $S_4$ and the emission of $PM_{10}$. $SO_2$ observations are used to constrain the $SO_2$ concentrations and emissions. $NO_2$ observations are used to update the NO and $NO_2$ concentrations and NO emissions. CO observations are used to constrain the CO concentrations and emissions, while $O_3$ observations are used to constrain the $O_3$ concentrations.

To clarify the details of the assimilation process and the treatment for the secondary formation, the text in Lines 171-177, Page 7, is added. The details of the updated chemical and emission species follow Peng et al. (2018), which is not include in the text.

"Since the secondary formation process can be captured by the WRF-Chem model, the impact of the secondary $PM_{2.5}$ is indirectly considered. The detailed updated state variables with the according observations follow Peng et al. (2018). The concentrations and emissions of $PM_{2.5}$, $NH_3$, and $PM_{2.5}$ precursors that have observations ($SO_2$ and NO), are updated by the observed quantities, respectively, but the VOC that are also $PM_{2.5}$ precursors are not updated due to the lack of direct and limited observations."

*2. Related to the first comment and also commented by the second reviewer. Is PM2.5 concentration also updated? Or just PM2.5 emission?*

Thank the reviewer for the insightful suggestion. In our reply to the previous comment, both $PM_{2.5}$ emissions and their corresponding concentrations are updated in the DA experiments.

*3. I seriously doubt the diurnal variations in PM2.5 emission (Fig. 6). Many studies assume that high emission rates during daytime (working hours) and low emission rates during nighttime as in Fig. 8. I think the highest emission rate in the morning in Fig. 6*

*is attributable to 1) high emission during rush hours and 2) shallow boundary layer. In other words, the diurnal variations in PM2.5 emissions estimated in this study do include the effects of time-varying boundary layer (and height). So, the effects of boundary layer are not separated from the emission estimates. We would expect high emission rates in the afternoon (working hours) and also during the late afternoon (evening rush hours). Because boundary layer height is generally highest in the late afternoon, the estimated emission rates in the late afternoon are too low (Fig. 6). I think monthly emission estimates or yearly estimates would be fine because the diurnally varying boundary layer is all averaged out at monthly and yearly time scales. To verify this, you can take a closer look at emission rates near industrial complex where diurnal variations in emissions are expected to be small (e.g., power plants, steel and cement companies ...). I understand the horizontal grid of 45 km is too coarse to examine this, but I expect that there are some regions where many factories are concentrated.*

Thank the reviewer for the insightful comment and suggestion. We performed an observing system simulation experiment (OSSE) to investigate the effects of time-varying boundary layer. A nature run is first conducted from 0000 UTC 25 December 2015 to 0000 UTC 2 February 2016, forced by the time-invariant source emissions PR2010 (the true emission). Synthetic observations of the six conventional air pollutant concentrations (i.e., $PM_{10}$, $PM_{2.5}$, $SO_2$, $NO_2$, $O_3$, and CO) are generated from the natural run. Hourly synthetic observations are created from 0000 UTC 29 December 2015 to 0006 UTC 1 February 2016, by interpolating the gridded true surface concentrations to the chemical observation locations with additive random errors of $N(0,R)$. $R$ is the observation error variance, which is calculated by the formula in Elbern et al. (2007). Outputs from the first four days of the natural run are excluded to avoid the transient effect. Then the prior emissions are generated by $F^{pr} = (1.8 + \delta(x, y, z, t))F^{tr}$, where $F^{tr}$ is the true emission, $\delta$ is a random number sampled from the normal distribution $N(0,1)$ (Peng et al. 2015). Ensemble data assimilation experiments are conducted from 0000 UTC 29 December to 0006 UTC 1 February 2016. Outputs from the first two days of the OSSE are excluded due to the spin-up.

Figure S1 presents the monthly mean diurnal variations of $PM_{2.5}$ emission fraction from the OSSE. Please note that the magnitude of posterior $PM_{2.5}$ emission is closer to the

true emission than the prior. Figure S1 shows that a little larger estimated $PM_{2.5}$ emission fractions occurred in the morning and smaller estimated $PM_{2.5}$ emission fractions occurred in the afternoon, comparing to the time-invariant true emission. This result is consistent with the reviewer's expectation. But the diurnal variations of $PM_{2.5}$ emission fractions caused by the boundary layer are not as strong as that caused by the emission itself (original Figure 6).

Since we have hourly assimilated observations to simultaneously update the chemical concentrations and source emissions, the impacts of time-varying boundary layer on the posterior $PM_{2.5}$ emissions are limited. Nevertheless, the influences of time-varying boundary layer are still important to $PM_{2.5}$ emission estimates. Thus, the diurnal variations of $PM_{2.5}$ emission fractions are updated to exclude the impacts of time-varying boundary layer, following the reviewer's suggestion. We modified Figures 7 and 8, and Table 2, to reflect the revision.

Compared to the diurnal variations of $PM_{2.5}$ emission fractions estimated based on diurnal variation profiles from US and EU (Wang et al., 2010), the estimated $PM_{2.5}$ emission fractions are 1.25% larger during the evening, which greatly changes the diurnal variations of DEPE. In fact, the smaller evening peaks of Wang et al. (2010) occurred in November, December, January, February and March, while they are almost indistinct from April to October, similar to that from DEPE.

The text in Lines 271-281, Page 10, and Appendix is added. The manuscript is updated by considering the effects of the boundary layer.

Elbern, H., Strunk, A., Schmidt, H., and Talagrand, O.: Emission rate and chemical state estimation by 4-dimensional variational inversion, Atmos. Chem. Phys., 7, 3749 – 3769, https://doi.org/10.5194/acp-7-3749-2007, 2007.

*4. Related to the effects of meteorology (or boundary layer), I would suggest some extra experiments (also related to the first reviewer's 4th minor comment asking "weather effects"). Let's fix anthropogenic emissions all the time, and only consider time-varying meteorology. Assume the observations that will be assimilated here are the model outputs with the same emissions but time-varying meteorology (not real observations). Then, assimilate these fake observations (actually model outputs) and estimate emissions. Would your estimated emissions be almost identical to the prior emissions that are fixed with time? I'm curious if your estimated emissions depend on / are*

*influenced by meteorology. A month-long simulation would be enough for this type of simulation.*

Thank the reviewer for the insightful suggestion. In our reply to the previous comment, an OSSE was performed to investigate the impacts of time-varying boundary layer on the diurnal variations of $PM_{2.5}$ emission fractions. The results indicate that the diurnal variations of the estimated $PM_{2.5}$ emission fractions from the OSSE do deviate from the true time-invariant emissions, although the magnitudes are not as large as those caused by emission itself. The diurnal variations of $PM_{2.5}$ emission fractions are updated by excluding the effects of time-varying boundary layer, following the reviewer's suggestion.

*5. line 306-307. Did you mean that emissions in 2019 were higher than those in 2020? I think in 2020 there were few firework activities due to the lockdown. If this is true, the color for BTH and SCR in Fig 11e should be blue (lower emissions in 2020 than in 2019). If not, please clarify. In addition, some studies highlighted that the PM2.5 concentration during Feb. 2020 is due to unfavorable meteorological condition in the BTH region (Sulaymon et al. 2021). Le et al. (2020) also showed that for the severe haze in northern China during the lockdown is due to 1) anomalously high humidity that promoted aerosol heterogeneous chemistry, 2) stagnant airflow 3) uninterrupted emissions from power plants and petrochemical facilities, and 4) secondary aerosol formation associated with increased ozone.*

Thank the reviewer for pointing out this mistake. The massive emissions from firework burning on the Chinese New Year Eve of year 2019 occurred on February 4, 2019. But in the following 7-day spring holidays, the smaller $PM_{2.5}$ emission rates are obtained (Figure 10). The possible reason of the increased PM2.5 emissions for BTH and SCR in 2020 is the long national vocation of spring holiday of year 2019. We changed the sentence in Line 356, Page 13.

We agree with the review that although there were significant reductions of $PM_{2.5}$ emissions over the central and northern China in February 2020, a severe air pollution event occurred over the north China in early February 2020. Previous studies have shown that the factors influencing the severe air pollution event include the still intensive emissions from industrial, power and residential, unfavorable meteorological condition, anomalously high humidity that promoted aerosol heterogeneous chemistry,

and secondary aerosol formation associated with increased atmosphere oxidants (Le et al. 2020; Sulaymon et al. 2021; Li et al., 2021). This discussion is added in the text at 363-369, Page 13.

*6. Constant emissions can be misleading (line 149, line 183, line 233…). Did you mean time invariant emissions? That is, emission rates do not vary with time at all at a grid cell. If so, I would recommend saying time-invariant emissions or constant emissions with time because the constant emissions can be interpreted as spatially homogeneous emissions.*

Thank the reviewer for the insightful comment. We mean time invariant emissions and we have changed *constant* emissions to *time-invariant* emissions in the revised manuscript.

*7. Figure 9. The x-axis label should be date, not time, right? And recommend representing mm/dd, format.*

The x-axis label is Date. We have changed this figure.

*References*
*Sulaymon et al. 2021. Persistent high PM2.5 pollution driven by unfavorable meteorological conditions during the COVID-19 lockdown period in the Beijing-Tianjin-Hebei region, China. Environmental Research. https://doi.org/10.1016/j.envres.2021.111186*
*Le et al. 2020. Unexpected air pollution with marked emission reductions during the COVID-19 outbreak in China, Science. DOI: 10.1126/science.abb7431*

---

## Author Response (AR2)

**Response to Reviewer #1**

Thank the reviewer for the insightful and detailed comments and suggestions, which helped to significantly improve the manuscript.

The reviewer's comments are shown in *blue italics* with the author responses in black.

*The authors made necessary revisions that address many of my concerns in the first round of review. I will outline several remaining comments below for the authors and the editor to consider, and I support the publication of the paper if they can be addressed.*

*1) Line 174: what observations are used to constrain NH3?*

Thank the reviewer for the thoughtful comment. $PM_{2.5}$ observations are used to constrain $NH_3$. This sentence is revised as (Lines 174-177):

"The concentrations and emissions of $PM_{2.5}$ and $PM_{2.5}$ precursors ($SO_2$ and $NO$) that have observations are updated by the observed quantities, respectively. Besides, $NH_3$ concentrations and emissions are constrained by $PM_{2.5}$ observations, however, the VOC that are also $PM_{2.5}$ precursors are not updated due to the lack of direct and limited observations."

*2) Section 5: My suggestion is this section can be cut or at least significantly shortened. The revised text reads to me that it is impossible to attribute how much of the derived PM2.5 emission changes between 2019 and 2020 are due to COVID. The absolute changes (Figure 10) are small. These changes are not strong enough relative to the changes in previous years (e.g., Table 1), so that COVID lockdown cannot be clearly attributed as a significant cause.*

*If the authors choose to keep this section in a shorter version, please discuss why the overall changes are so small relative to bottom-up studies (e.g., Zheng et al., 10.5194/essd-13-2895-2021, 2021). This might be true since COVID lockdown might increase residential emissions while reducing traffic emissions. But overall, the results provide little information towards a thorough explanation.*

Thank the reviewer for the valuable and thoughtful comment. We agree with the

reviewer that the changes between 2019 and 2020 are not strong enough relative to the changes in previous years for the whole year over the whole country (Figure 10, Table 1). However, although similar emission reductions and emission trends are obtained from the bottom-up technique (Zheng et al., 2021), the reduction amount and ratio from the bottom-up technique are larger than those estimated from DEPE (Figure 10 and Table 1). This is possibly due to significant reductions of $PM_{2.5}$ emission from the residential sector as in the bottom-up technique (Zheng et al., 2021), however, $PM_{2.5}$ emissions from the residential sector might not significantly changed around the COVID outbreak.

The abrupt changes of $PM_{2.5}$ emissions during the initial stage of COVID-19 in China provide a natural case study to validate the ability of the dynamic-based data assimilation method to obtain high temporal-resolution $PM_{2.5}$ emission estimates. Therefore, we discussed the assimilated results in detail in Section 5. We add this discussion in the text in Lines 334-336, 351-356.

*3) Line 363-369: How would these factors affect PM2.5 emissions or the top-down inversion?*

Thank the reviewer for the valuable comment. We delete this paragraph since it has less relation with the subject of the manuscript.

**Response to Reviewer #3**

We thank reviewer for his thoughtful comments and suggestions that have helped to significantly improve this manuscript.

The reviewer's comments are shown in *blue italics* with the author responses in black.

*Thanks for addressing my comments. The revised manuscript is overall well prepared. I have a couple of minor/technical comments.*
*1. Regarding the OSSE, did you run your model over the entire period and then substract the effect of boundary layer (or meteorology) over the entire period? Or, did you apply the result of the short-term period simulation (Jan 2016) to the entire period? If the former is the case, then it should be fine. But, the latter is the case, please specifically mention in your revised manuscript that the effect of meteorology is based on the short-term period simulation. This is because the effect of boundary layer or meteorology can be largely different depending on seasons or years.*

Thank the reviewer for the thoughtful comment. The OSSE is performed from 0000 UTC 29 December to 0006 UTC 1 February 2016. The effect of boundary layer is roughly estimated based on the assimilation results from 1 to 31 January 2016. We agree with the reviewer that the influences of boundary layer could strongly vary with seasons or years and mentioned these in the revised versions in Lines 275, 285-286.

*2. The caption of Fig. 6 is not very clear. I would recommend adding titles above Fig. 6a and 6b, and saying that Dec 2017 - Nov 2017, Dec 2018 - Nov 2017, respectively. Also, please revise the caption to clearly indicate "compared to what".*

Thank the reviewer for the thoughtful comment. We have added titles above Fig. 6a and 6b. And we also revised the caption in Fig. 6.

*3. Fig. 11. Description of (a) is missing.*

Thank the reviewer for the valuable comment. We have revised the caption in Fig. 11.